Manuscript prepared for Atmos. Meas. Tech. Discuss.
with version 4.1 of the LATEX class copernicus_discussions.cls.
Date: 27 February 2018

# Retrieval of Total Water Vapour in the Arctic Using Microwave Humidity Sounders

**Raul Cristian Scarlat**[1]**, Christian Melsheimer**[1] **and Georg Heygster**[1]

[1]Institute of Environmental Physics, University of Bremen, Germany

Correspondence to: Raul Cristian Scarlat
(rauls@iup.physik.uni-bremen.de)

## Abstract

Quantitative retrievals of atmospheric water vapour in the Arctic are faced with numerous challenges because of the particular climate characteristics of this area. Here, we attempt to build upon the work of Melsheimer and Heygster (2008) to retrieve total atmospheric water vapour
(TWV) in the Arctic from satellite microwave radiometers. While the above-mentioned algorithm deals primarily with the ice-covered central Arctic, with this work we are aiming to extend the coverage to low ice cover and ice-free areas. By using modeled values for the microwave emissivity of the ice-free sea surface, we develop two sub-algorithms using different sets of channels that deal solely with open ocean areas. The new algorithm extends the spatial cover-
age of the retrieval throughout the year but especially in the warmer months when higher TWV values are frequent. The high TWV measurements over both sea ice and open water surfaces are however connected with larger uncertainties as the retrieval values are close to the instrument saturation limits.

This approach allows us to apply the algorithm to regions where previously no data were
available and ensures a more consistent physical analysis of the satellite measurements by taking into account the contribution of the surface emissivity to the measured signal.

## 1  Introduction

Water vapour plays a crucial role within the complex system of our atmosphere. It transports energy from the warmer lower latitudes to higher ones influencing global weather patterns,
plays a big part in trapping infrared (IR) radiation, and is highly variable throughout the planetary atmosphere (Le and Gallus Jr, 2012). The water vapour content of air is regulated through the processes of condensation, evaporation and, since the advent of life on this planet, transpiration. These phase changes provide the mechanisms through which water vapour influences atmospheric temperature by adding or removing energy from the air. Through evaporation, en-
ergy is stored as latent heat within the water molecules that break away from the liquid as gas, thus leading to a cooling effect in its immediate vicinity. The reverse of this process releases this

latent heat through condensation and provides energy, e.g., for the development of thunderstorm cells.

One of the main characteristics of atmospheric water vapour is its high variability (Trenberth et al., 2005), both in terms of spatial location and temporal evolution. Because of this variability water vapour can be used as an atmospheric tracer that indicates general atmospheric circulation as it accompanies the horizontally moving air masses while its phase changes indicate the location of upwelling or downwelling currents. The average lifetime of a molecule of water in the atmosphere is 10-12 days, during which it can go through many phase changes.

Water vapour is one of the major greenhouse gases in the atmosphere. As such, it is important to monitor the variability of water vapour considering the anthropogenic increase of other greenhouse gases (Solomon et al., 2010). In the context of global warming the atmosphere's load capacity for water vapour increases and therefore its contribution as the most important greenhouse effect warrants precise monitoring. Model data indicate that this positive feedback loop would increase the sensitivity of surface temperatures to carbon dioxide concentrations by a factor of two (Held and Soden, 2000) without taking into consideration other possible feedback processes. However, this matter is still debated, as the atmospheric storage of water vapour is not understood well enough to warrant definitive conclusions. In the context of climate change, the importance of water vapour as a greenhouse gas and the positive feedback loop with which it is associated with the global temperature make it necessary to obtain accurate data on atmospheric water vapour load on a global scale.

For the purpose of measurement, we quantify the atmospheric water vapour load as a vertically integrated mass over a column of air with the base area of 1 $m^2$ and name it as total water vapour or TWV for short. It is the parameter that we will be discussing throughout this paper when referring to measured atmospheric water vapour content.

A reliable method to retrieve atmospheric water vapour content is by using balloon-borne radiosondes. This is an accurate method which provides good vertically resolved measurements. However, these are only point measurements and thus only of local significance (Hurst et al., 2011). Ground-based retrievals such as fixed radiometers and GPS-based retrievals achieve a lower vertical resolution but are considered feasible for monitoring purposes in regions with

a good density of ground stations (Das et al., 2014). However, only satellite measurements fulfil the global coverage requirements of modern numerical weather prediction. Because of the strong absorption properties of water vapour in the infrared and microwave range, suitable space-borne instruments can ensure a complete global coverage of water vapour retrievals (Miao, 1998; Szczodrak et al., 2005; Bobylev et al., 2010).

Radiosonde retrieval of total water vapour in the Arctic is not sufficient because of the scarcity of weather stations in this area. The inhospitable environment presents further challenges to obtain a satisfactory coverage from ground data (Serreze and Hurst, 2000). Satellite retrievals also face a number of obstacles. Infrared measurements are hampered by cloud cover and, while microwave radiometer measurements are a viable option, an incomplete understanding of sea ice emissivity properties challenges retrieval efforts in this area.

One important step towards achieving a satisfying retrieval of TWV in the polar regions came from the work of (Miao et al., 2001). It uses the Special Sensor Microwave/ Temperature-2 (SSM/T2) humidity sounder and was designed to work in the Antarctic and will be referred throughout this paper as the Antarctic algorithm, to differentiate it from the other discussed algorithms which were developed for the Arctic. The key concept of this method is the use of several microwave channels with similar surface emissivity but different water vapour absorption behaviour. These are the three channels near the 183.31 GHz water absorption line ($183.31\pm1,\pm3,\pm7$ GHz), which together with the channel at the 150 GHz window frequency allow retrieval of TWV values up to about $7\,\mathrm{kg/m^2}$. Above this value two of the 183.31 GHz band channels become saturated and the sensor fails to 'see' down to the ground anymore. This limit is sufficient for the central Arctic region for most of the year and it shows good agreement with retrievals based on the analysis of GPS signals taken at stations around Antarctica (Vey et al., 2004). However, because of the limited TWV retrieval range, this method alone cannot ensure a year long monitoring of TWV values (Selbach et al., 2003).

The Antarctic algorithm (Miao et al., 2001) worked independently of the surface type by assuming the same surface emissivity for all channel frequencies used. While this is a valid assumption for the case when the three 183.31 GHz band channels are used as the surface emissivities are very close to each other, this is a poor assumption when using a triplet that

includes the 150 GHz channel (Wang et al., 2001) which has a different surface emissivity from the 183.31 GHz bands. The emerging errors from the emissivity differences were deemed acceptable as a trade-off for extending the retrieval range from 1.5-2 kg/m$^2$ (when using all three band channels) up to 7 kg/m$^2$ (when using the 183.31±3 and 183.31±7 GHz together with the 150 GHz channel). To improve on this performance the algorithm developed by Melsheimer and Heygster (2008) extends the TWV retrieval range over sea ice by including the 89 GHz channel into the retrieval. Using the triplet of 183.31±7 GHz, 150 GHz and the 89 GHz channels allows the retrieval to function up to the saturation limit of the 183.31±7 GHz channel.

By using the 89 GHz channel, the difference in surface emissivity for the different frequencies becomes too great and the equal emissivity assumption has to be dropped. In order to increase the retrieval range, some information about the emissivity of the ground surface is necessary. Because the surface emissivity of sea ice is very different from that of water, the retrieval algorithm needs to treat each surface type differently. In Melsheimer and Heygster (2008) the priority was to first extend the TWV retrieval range over sea ice as this was a new capability while TWV retrieval algorithms that can function over open water were already operational. After implementing emissivity information about the sea ice surface, the algorithm can retrieve up to 15 kg/m$^2$ with an error of ≈3 kg/m$^2$ above sea ice-covered areas. For values below 7 kg/m$^2$, this algorithm uses the same retrieval mechanism as the Antarctic algorithm. While providing a boost to the retrieval range of TWV in the Arctic, this method proved the feasibility of using ground emissivity information to achieve passive microwave TWV retrieval over different surface types. In this paper we use the well-understood microwave emissivity of the ice-free sea surface to develop two sub-algorithms for the Melsheimer and Heygster (2008) method that deals solely with open ocean areas. This approach allows for application of the extended range algorithm to regions where previously no data were retrievable because of the proximity of sea ice that prevents open water TWV retrieval algorithms to work or because of the relatively high TWV value that could not be retrieved by the original method over partially ice covered regions. Throughout the rest of this paper we will refer to the Melsheimer and Heygster (2008) algorithm as the original method on which the development of the new algorithm was based on. Because the new algorithm adds additional capabilities, uses modified retrieval equations and increases

the complexity of the retrieval we believe the two methods are distinct enough to be compared with each other as stand alone algorithms that use the same working principle.

In Section 2 the basic TWV retrieval in the case of equal surface emissivity is discussed. Following this, we introduce the subsequent extensions to the algorithm starting with the first extension for retrieval over sea ice done by Melsheimer and Heygster (2008) and continuing with the open water extension which is the topic of this current study. Section 3 contains the results of comparing the original retrieval with the new method as well as an intercomparison between the new method and two other retrieval products. In Section 4 the conclusions are presented.

## 2 Methods

### 2.1 Radiative transfer equation

As with many other passive microwave retrieval techniques, the algorithm uses a radiative transfer equation to interpret the data from a humidity sounder such as AMSU-B (Advanced Microwave Sounding Unit-B) or MHS (Microwave Humidity Sounder) on board the NOAA (National Oceanic and Atmospheric Administration) 17,18 satellites. Although the method has been tested using the newer MHS instrument data in order to ensure continuity of operation all results presented in this work are based on AMSU-B measurements. A down-looking microwave radiometer, such as the AMSU-B humidity sounder, will measure upward radiances at the top of the atmosphere. They can be expressed as brightness temperatures of the Earth's atmosphere. Using the simplified radiative transfer equation from Guissard and Sobiesky (1994), we express the radiance measured by the instruments as the brightness temperature

$$T_b(\theta) = m_p T_s - (T_0 - T_c)(1 - \epsilon)e^{-2\tau sec\theta}. \tag{1}$$

Here $\theta$ is the off-nadir viewing angle of the satellite, $m_p$ is a correction factor that accounts for the deviation from an isothermal atmosphere and the difference between surface and air tem-

perature. $T_s$ is the surface temperature, $T_c$ the brightness temperature of the cosmic background contribution, and $T_0$ is the ground level atmospheric temperature. $\epsilon$ is the surface emissivity, while $\tau$ is the atmospheric opacity. The challenging term here is the correction factor $m_p$ which has to be approximated. For an ideal case of an isothermal atmosphere, the ground being a specular reflector and the surface skin temperature being equal to the ground level atmospheric temperature, $m_p$ would be equal to unity. The Melsheimer and Heygster (2008) algorithm assumes the ground to be a specular reflector which is a sufficient approximation for remote sensing applications in the microwave domain according to Hewison and English (1999).

## 2.2 The basic idea of TWV retrieval with equal surface emissivity assumption

The entire path leading from the radiative transfer equation and up to the final atmospheric water vapour $W$ retrieval equation has been covered in the initial Antarctic algorithm paper Miao et al. (2001) and in the subsequent Arctic extension paper Melsheimer and Heygster (2008). We will review it here because the basic mechanism remains the same and is incorporated in the low TWV retrieval component of our final method.

As long as no channel is saturated, i.e., the channel signal still comes from the entire atmospheric column down to the surface and not just the upper layers, all channels "see" the ground and the surface contribution is the same for all three measurements. The water vapour absorption will be different for the three channel frequencies. If $i, j, k$ are the channel indices ordered by decreasing difference to the absorption line maximum then the mass absorption coefficients $\kappa$ will be $\kappa_i < \kappa_j < \kappa_k$. To cover the whole retrieval range, the Antarctic algorithm used two channel triplets

   i) 183.31±7, 183.31±3, and 183.31±1 GHz (AMSU-B channels 20, 19, 18); or

   ii) 150, 183.31±7, and 183.31±3 GHz (AMSU-B channels 17, 20, 19).

For the first channel triplet the assumption of equal emissivity is fulfilled because the three frequencies are so close to each other. For the second triplet, the same assumption is still used although the difference in frequencies is greater and some inaccuracy is introduced in the retrieval. In Miao (1998), it is argued that using this assumption for the second channel triplet represents a small error source when compared to other ones. Quantitatively it has been shown

(Wang et al., 2001; Selbach, 2003; Selbach et al., 2003) that using the same emissivity assumption while including the 150 GHz channel will cause a positive bias of up to $0.5\,\text{kg/m}^2$ depending on the type of surface. We can simplify the expression of brightness temperature given in Eq. (1) by taking the difference of two brightness temperatures measured at two different channels $i, j$, so that we get

$$\Delta T_{ij} \equiv T_{b,i} - T_{b,j} = (T_0 - T_c)(1 - \epsilon)(e^{-2\tau_j sec\theta} - e^{-2\tau_i sec\theta}) + b_{ij}. \tag{2}$$

To account for the differences in the $m_p$ term, the bias term $b_{ij}$ was introduced.

$$b_{ij} = T_s(m_{p,i} - m_{p,j}). \tag{3}$$

As shown in Melsheimer and Heygster (2008) - Appendix II, a good approximation for this term is

$$b_{ij} \approx \int\limits_{0}^{\infty} [e^{-\tau_j(z,\infty)sec\theta} - e^{-\tau_i(z,\infty)sec\theta}]\frac{dT(z)}{dz}dz. \tag{4}$$

Here, $T(z)$ stands for the temperature profile of the atmosphere with height $z$. To find the relationship between the measured brightness temperature and the water vapour absorption that does not depend on any other surface parameter we require the third brightness temperature measured in channel $k$. With this, a pair of brightness temperature differences is available from which the ratio will be

$$\eta_c \equiv \frac{\Delta T_{ij} - b_{ij}}{\Delta T_{jk} - b_{jk}} = \frac{e^{-2\tau_i sec\theta} - e^{-2\tau_j sec\theta}}{e^{-2\tau_j sec\theta} - e^{-2\tau_k sec\theta}}. \tag{5}$$

By using the two brightness temperature differences between three brightness temperatures and taking the ratio of these differences all terms that depend on surface parameters have been simplified and now we have a direct relationship between the measured brightness temperatures and atmospheric opacity due to water vapour absorption. Following the naming convention in

Melsheimer and Heygster (2008), we call the left hand side of Eq. (5) the ratio of compensated brightness temperatures, $\eta_c$ (containing the correction terms $b_{ij}$ and $b_{jk}$). $\eta_c$ is independent of any surface contribution, and only influenced by the atmospheric opacity terms $\tau_{(i,j,k)}$ at the three channel frequencies. These atmospheric opacity terms in turn are functions of the amount of absorption by water vapour and oxygen and can be expressed as

$$\tau_i = \kappa_{vapour,i} W + \tau_{oxygen,i}, \tag{6}$$

where $\kappa_{vapour,i}$ is the water vapour mass absorption coefficient at channel $i$, $\tau_{oxygen,i}$ represents the oxygen contribution to the atmospheric attenuation at channel $i$, and $W$ is the water vapour load. For the band channels around the 183.31 GHz frequency, the contribution of water vapour to the absorption is much stronger than for oxygen so that the $\tau_{oxygen,i}$ term will be neglected henceforth.

The aim is to have a direct connection between the ratio of brightness temperature and the water vapour content $W$. Using the approximation introduced by Miao (1998), the difference of exponentials can be transformed into a product of a linear and an exponential function so that eventually we get

$$\eta_c = \exp[B_0 + B_1 W \sec\theta + B_2 (W \sec\theta)^2]. \tag{7}$$

Here $B_0$, $B_1$, and $B_2$ depend on the mass absorption coefficients $k$ for the three channels and are called bias parameters. Compared to the first two terms under the exponent the quadratic term is negligible small so that the logarithm of Eq. (7) becomes

$$\log \eta_c = B_0 + B_1 W \sec\theta. \tag{8}$$

The final retrieval equation for water vapour content $W$ is then

$$W \sec\theta = C_0 + C_1 \log \eta_c, \tag{9}$$

where $C_0 = -\frac{B_0}{B_1}$ and $C_1 = \frac{1}{B_1}$ characterizing the atmospheric attenuation at the channel frequencies used. These calibration parameters are determined from simulated brightness temperatures based on radiosonde profiles. The simulations were run using the ARTS (Atmospheric

Radiative Transfer Simulator) radiative transfer model (Eriksson et al., 2011) which used as input radiosonde data collected from 29 coastal or island WMO (World Meteorological Organization) stations in the Arctic. The time period for the radiosonde measurements used is between 1996 and 2002.

By replacing the form of $\eta_c$ from Eq. (7) in the ratio of brightness temperature differences from Eq. (5) we obtain the linear relationship between $\Delta T_{ij}$ and $\Delta T_{jk}$

$$\Delta T_{ij}(\epsilon) = b_{ij} + \eta_c(W)(\Delta T_{jk}(\epsilon) - b_{jk}). \tag{10}$$

The brightness temperature differences depend on the surface emissivity $\epsilon$ while $\eta_c$ only depends on $W$. In a $\Delta T_{ij}$ vs. $\Delta T_{jk}$ scatter plot with constant $W$ and for varying $\epsilon$, Eq. (10)
describes a straight line of slope $\eta_c(W)$ that runs through the point $(b_{ij}, b_{jk})$. Because the two bias parameters vary only weakly with $W$ and $\eta$, all lines obtained for different $W$ values will cross or pass very near to one single point $F(F_{jk}, F_{ij})$, called focal point (Miao et al., 2001). To find its coordinates, brightness temperature simulations were run for eleven discrete values of $\epsilon$. As for the calibration parameters described above, the simulations are run with the ARTS
radiative transfer model using Arctic radiosonde profiles with realistic $W$ values as input. For all simulations, the surface temperature equals the ground level atmospheric temperature. For each constant $W$ value, a line is fitted to the the points in the $\Delta T_{ij}$ vs. $\Delta T_{jk}$ scatter plot. The point of least square distance from all lines will be called the focal point $F$. By finding the focal point coordinates we have the relationship between the simulated brightness temperature differ-
ences and the $W$ values and so fit Eq. (9), which allows us to determine the constant calibration parameters $C_0$ and $C_1$. With this method a total of four parameters, two focal point coordinates and two atmospheric calibration parameters are derived through the regression fit.

The principal problem with the Antarctic algorithm was that the sensitive band channels around the 183.31 GHz frequency will reach saturation with relatively low amounts of atmo-
spheric water vapour (Selbach, 2003). This means that after crossing a certain threshold value for $W$, the brightness temperature $T_b$ does not vary with increasing $W$. Therefore, when one channel reaches saturation, it can no longer be used in the retrieval triplet for higher $W$ values. For the first channel triplet (183.31 $\pm$7, 183.31$\pm$3, and 183.31$\pm$1 GHz), the first channel

(AMSU-B channel 18 at 183.31±1 GHz) reaches saturation at 1.5 kg/m². To achieve a practical $W$ retrieval range, the channel triplet (17, 20, 19, i.e. 150, 183.31 ± 7, and 183.31 ± 3 GHz) is used for values higher than 1.5 kg/m² and can function up to 7 kg/m² when channel 19 reaches saturation. As a practical test for when the algorithm should switch between the two channel triplets, the saturation point for a given channel $k$, as defined in Miao et al. (2001), is the $W$ threshold value after which $T_{b,j} \leq T_{b,k}$, or simply

$$T_{b,j} - T_{b,k} \geq 0. \tag{11}$$

This test is based on the threshold at which the brightness temperature of a given channel is no longer increasing with increasing $W$. This threshold represents the point at which the brightness temperature levels off and then starts to decrease again with increasing $W$. This happens as the signal that reaches the instrument no longer comes from the whole atmospheric column down to the ground but only from the colder, upper part of the atmosphere. In the original version of the Arctic algorithm (Melsheimer and Heygster, 2008), in order to extend the retrieval range, the above condition has been relaxed. The saturation cut-off temperature, $T_{bj} - T_{b,k} \geq 0$ has been modified to $F_{20,19}$ ( $T_{bj} - T_{b,k} \geq F_{20,19}$), with $F_{20,19}$ being a few Kelvins. This modification translates into an increase in the retrieval range by about 1 kg/m² at the expense of increased error as the channel approaches its saturation limit. If the difference $\Delta T_{jk} - F_{jk}$ is smaller than $-10 K$ the corresponding error for the second channel triplet (17, 20, 19) is below 0.4 kg/m² for the retrieval range 1.5 – 7 kg/m². For the first channel triplet (20, 19, 18) which has the narrow retrieval range of 0 – 1.5 kg/m² the error is below 0.2 kg/m². This particular issue of the relaxed conditions will be addressed in the final algorithm (Section 2.7). These specific cases where the Melsheimer and Heygster (2008) algorithm would only retrieve data under the relaxed condition scenario were found to be mostly open water regions where the equal emissivity assumption failed. The new components that deal exclusively with retrieval over open water use the condition in Eq. (11).

## 2.3 Extending the TWV retrieval range

Normally, for TWV values above $7\,\mathrm{kg/m^2}$, saturation occurs at channel 19 (183.3±3 GHz). To extend the retrieval range above this threshold, a new channel is necessary to take its place in the triplet, which means that a new set of assumptions has to be made about the surface emissivity influence. Now, the three channels $i, j, k$ represent AMSU-B channels 16, 17 and 20 (89,150 and 183.31±7 GHz ). Because channel 16 is so far apart from the other two, we can no longer assume that it has the same surface emissivity as the others. Therefore, we will have $\epsilon_i \neq \epsilon_j$ for the new channel $i$ and $\epsilon_j = \epsilon_k$ is the approximation used as before.

If we consider that we have two channels with different surface emissivities, the brightness temperature difference becomes

$$\Delta T_{ij} \equiv T_{b,i} - T_{b,j}$$
$$\Delta T_{ij} = (T_0 - T_c)(r_j e^{-2\tau_j sec\theta} - r_i e^{-2\tau_i sec\theta}) + b_{ij}, \tag{12}$$

where $r$ is the ground reflectivity $(1 - \epsilon)$, and $b_{ij}$ is the same as in Eq. (4) because it does not depend on the surface emissivity $\epsilon$. The corresponding compensated ratio of brightness temperature differences is

$$\eta_c = \frac{\Delta T_{ij} - b_{ij}}{\Delta T_{jk} - b_{jk}} = \frac{r_i e^{-2\tau_i sec\theta} - r_j e^{-2\tau_j sec\theta}}{r_j(e^{-2\tau_j sec\theta} - e^{-2\tau_k sec\theta})}. \tag{13}$$

Rearranging terms to resemble the original form in Eq. (5) we get

$$\eta_c = \frac{r_i(e^{-2\tau_i sec\theta} - e^{-2\tau_j sec\theta})}{r_j(e^{-2\tau_j sec\theta} - e^{-2\tau_k sec\theta})} - \left(1 - \frac{r_i}{r_j}\right)\left(\frac{e^{-2\tau_j sec\theta}}{e^{-2\tau_j sec\theta} - e^{-2\tau_k sec\theta}}\right). \tag{14}$$

After approximating the difference in exponentials as before, the compensated ratio of brightness temperature differences becomes

$$\eta_c = \frac{r_i}{r_j} \exp[B_0 + B_1 W \sec\theta + B_2(W \sec\theta)^2] - \left(1 - \frac{r_i}{r_j}\right) C(\tau_j, \tau_k), \tag{15}$$

where

$$C(\tau_j, \tau_k) = \frac{e^{-2\tau_j \sec\theta}}{e^{-2\tau_j \sec\theta} - e^{-2\tau_k \sec\theta}}$$

is a slowly varying function that depends only on the atmospheric absorption factors. In order to obtain a simple linear relationship between the compensated brightness temperature difference and water vapour load $W$ we rearrange the above equation into

$$\log \eta_{c\prime} = B_0 + B_1 W \sec\theta + B_2 (W \sec\theta)^2. \tag{16}$$

The modified ratio $\eta_{c\prime}$ includes the terms depending on the reflectivities and the $C(\tau_j, \tau_k)$ function

$$\eta_{c\prime} = \frac{r_i}{r_j}[\eta_c + C(\tau_j, \tau_k)] - C(\tau_j, \tau_k) \tag{17}$$

The final retrieval equation for $W$ is obtained after eliminating the negligible quadratic term

$$W \sec\theta = C_0 + C_1 \log \eta_{c\prime}. \tag{18}$$

The difference between $\eta_{c\prime}$ and $\eta_c$ is that the former depends on the surface emissivity through the reflectivities ratio $\frac{r_i}{r_j}$ while the latter is surface independent. To enable retrieval using Eq. (18), more information is needed about the behaviour of the surface emissivities at 150 and 89 GHz. Direct information about the surface emissivity for every satellite footprint is not available so we need to parametrize the emissivity and obtain a constant reflectivity ratio that would only roughly depend on the surface type ocean/ice/ land. Identifying the major surface types in the Arctic is another task that has to be integrated into the algorithm. Because at the time no ocean surface emissivity information was readily available and a proof of concept was needed first over regions with low enough TWV, the Melsheimer and Heygster (2008) algorithm extension was adapted only for sea ice surfaces. The sea ice surface emissivity data was obtained from the Surface Emissivities in Polar Regions-Polar Experiment (SEPOR/POLEX measurement campaign in 2001). This campaign used an aircraft-mounted instrument, the Microwave Airborne

Radiometer Scanning System (MARSS), which possesses two microwave channels of frequencies similar to those required for the algorithm extension. For AMSU-B channel 16 at 89 GHz, there exists the MARSS channel 88.992 GHz, and for AMSU-B channel 20 at 150 GHz there exists a corresponding MARSS channel at 157.075 GHz. This difference of 7 GHz does not
pose significant issues for the retrieval using the 150 GHz channel. The difference between measurements at 150 and 157 GHz is between $\pm\,0.01$ calculated from the in situ measurements (Selbach et al., 2003; Selbach, 2003), while the emissivity variability for the different ice types is greater than this difference. Because of this, the impact on the final retrieval is considered negligible.

To obtain the reflectivity ratio, the regression of $\epsilon_{89}$ as a function of $\epsilon_{150}$ was calculated

$$\epsilon_{89} = a + b\epsilon_{150}. \tag{19}$$

For the ratio of reflectivities to be constant it has to be independent of the variable emissivities. Because of this the regression was constrained so that $\epsilon_{89}$ ($\epsilon_{150} = 1$)$\approx 1$. The physical meaning of this is that the emissivity for the two channels cannot be greater than 1. Using the constraint
above means that $a + b \approx 1$ and so the reflectivity ratio only depends on the regression relationship coefficient $b$

$$\frac{r_{150}}{r_{89}} \approx \frac{1}{b}. \tag{20}$$

From the data points over sea ice, the following regression relationship was found by Melsheimer and Heygster (2008) for emissivity at 89 and 150 GHz

$$\epsilon_{89} = 0.1809 + 0.8192 \cdot \epsilon_{150} \tag{21}$$

By replacing the coefficient $b$ in Eq. (20) we have the reflectivity ratio

$$\frac{r_{150}}{r_{89}} = 1.22. \tag{22}$$

It is indicated by Melsheimer and Heygster (2008) that this is just a partial compensation for the contribution of surface emissivity. The SEPOR/POLEX measurements were made in the winter

season and therefore do not take into account the melt processes that take place in summer which can significantly alter the emissivity behaviour of the surface (Tonboe et al., 2003). Because other resources on the subject are sparse this was the only option to include the effects of surface emissivity into TWV retrieval.

Besides the two parameters $C_0$ and $C_1$ that account for the atmospheric conditions in the Arctic, the modified ratio of compensated brightness temperatures $\eta_c\prime$ requires the $C(\tau_j, \tau_k)$ term that depends on the atmospheric opacities and thus, directly on TWV. If one studies the behaviour of $C(\tau_j, \tau_k)$ with increasing TWV for values above $7\,kg/m^2$ the function varies only a little, and it can be approximated by a constant. According to Melsheimer and Heygster (2008),

a variation in $C(\tau_j, \tau_k)$ between 1.0 and 1.2 will result in a change of $C_0$ and $C_1$ in the third significant digit. In total we have the two focal point coordinates, the atmospheric parameters $C_0$ and $C_1$, and the slowly varying function approximated by the constant $C(\tau_j, \tau_k) \approx 1.1$. The set of four parameters is determined through regression by using simulated brightness temperatures and atmospheric data from radiosonde profiles as described in Section 2.2.

The weakness of the extended algorithm is its sensitivity to changes in the reflectivity ratio. In other words, for sea ice surfaces where the surface emissivity deviates within the uncertainty limits of $\sigma_{r_j/r_i} = 0.09$ (Melsheimer and Heygster, 2008) from the constant emissivities ratio used, the retrieval error can be as high as $3\,kg/m^2$.

Because of the specific channel triplet used by each sub-algorithm, the set of four calibration
parameters has to be determined for the low TWV, mid-TWV and extended-TWV cases. In the new algorithm, two extra sets of calibration parameters are required, for the mid-TWV over open water and extended-TWV over open water components.

## 2.4   Modifying the extended algorithm for use over open ocean

In Melsheimer and Heygster (2008) the possibility of using the same technique of incorporating
surface emissivity information for the purpose of using the extended range component over open water regions in addition to sea ice-covered areas was considered as a possible improvement but was not investigated further.

In order to determine the feasibility of this option, a suitable linear relationship between surface emissivities at channels 150 and 89 GHz is required. By reusing the retrieval equation and replacing the calibration coefficients and the ratio of reflectivities, a separate module for retrieving water vapour in the extended range only over open water can be implemented.

The ocean emissivity model FASTEM (FAST surface Emissivity Model for microwave frequencies) takes into account the characteristics of the AMSU-B instrument, sea surface temperature and roughness (Hocking et al., 2011). The parameter that was found to determine a strong variation in surface emissivity is the ocean surface roughness. Surface roughness in turn is determined by wind speeds. At the typical range of values encountered in the Arctic (8–

20 m/s), surface emissivity is determined mainly by wind speed. Figure 1 shows the behaviour of the ocean surface emissivities for the five channel frequencies of the AMSU-B instrument. Because the frequencies of the three band channels around 183.3 GHz are so close to each other, the corresponding emissivities are almost identical and thus represented by only one curve on the graph. Important to notice is the big difference between the curve for 89 GHz and the one

for 150 GHz, which illustrates why the assumption of equal emissivities cannot be sustained for these pairs of channels. Also the difference between the 183.3 GHz and the 150 GHz curves that is neglected by using the assumption of equal surface emissivity for the medium TWV retrieval range must be noted.

## 2.5 Ocean surface emissivity for the extended range 89,150,183.3±7 GHz triplet setup

Following the same method as for the extension over sea ice, through a linear regression between the ocean surface emissivity at 150 and at 89 GHz (panel a) of Fig. 2) we found the following linear relationship in the form of (10)

$$\epsilon_{89} = 1.2698 \cdot \epsilon_{150} - 0.2687. \tag{19}$$

For the emissivity of sea ice, studied for the first retrieval range extension, the constraint
$\epsilon_{89}(\epsilon_{150} = 1) \approx 1$ had to be imposed on the system in order to express the ratio of reflectivities

as a constant of the form shown in Eq. (7) independent of variable surface emissivity. Following the same logic, from the linear expression above we got the ratio of reflectivities

$$\frac{r_{150}}{r_{89}} = 0.7875.$$

Using this relationship, the calibration parameters $C_0$ and $C_1$ were also determined from
regression between $W$ from radiosonde profiles and simulated brightness temperatures as described in Section 2.2.

## 2.6 Ocean surface emissivity for the mid range 150, 183.3±7, and 183.3±3 GHz triplet setup

One of the error sources in the original algorithm was the assumption of equal surface emissivity
for the 150 GHz and the 183.3 GHz band channels. Over open ocean, the differences in surface emissivity at these frequencies can lead to a positive bias in the TWV retrieval. Following the same methodology as for the lowermost channel triplet (89, 150, 183.3±7 GHz), a linear relationship can be retrieved from simulated ocean surface emissivity data for the frequency triplet (150, 183.3±3, 183.3±7 GHz). From this, a reflectivity ratio can be obtained and used
in a modified retrieval equation. This modification leads to an improvement in the bias when retrieving in the TWV range 2-6 kg/m$^2$ over ice-free ocean surfaces.

Following the regression fit in panel b) Figure 2 we obtained the linear relationship for ocean surface emissivity at 150 and 183 GHz

$$\epsilon_{150} = 1.1022 \cdot \epsilon_{183} - 0.1028 \tag{24}$$

from which we obtain the ratio of reflectivities as

$$\frac{r_{183}}{r_{150}} = 0.9073.$$

In addition to the $C_0$ and $C_1$ parameters, the $C(\tau_j, \tau_k)$ function that depends on the atmospheric opacity is necessary for a retrieval when a different surface emissivity is considered

(Section 2.3). This function depends directly on TWV and it has been shown that above $7\,kg/m^2$ it is constant for the 89 and 150 GHz frequencies.

For the channels used in the mid-TWV range retrieval module, the function $C(\tau_j, \tau_k)$ behaves differently than for the extended-TWV channels. Between 2 and $6\,kg/m^2$ it drops rapidly from
1.4 down to 1.0 (Fig. 3), but it has been found (Melsheimer and Heygster, 2008) that changes on the order of 0.2 in $C(\tau_j, \tau_k)$ lead to differences in the third significant digit of the $C_0$ and $C_1$ parameters, which is small compared to other error contributions. In the process of modifying the mid-TWV algorithm, the $C(\tau_j, \tau_k)$ was recalculated for the 183±7 and the 150 GHz channels and set as a constant $C(\tau_j, \tau_k) \approx 1.15$ in the retrieval Eq. (18).

## 2.7 TWV algorithm synthesis

The final structure of the new algorithm comprises a collection of independent retrieval modules, each tuned to a different set of surface and atmospheric parameters. This structure can be viewed in Table 1 where the SSM/T2 Antarctic algorithm by Miao et al. (2001), the original AMSU-B Arctic retrieval algorithm by Melsheimer and Heygster (2008) and the new AMSU-B
algorithm are described. The main modules represent the three different channel triplets, low, mid and high, that are used in the different retrieval ranges of TWV. Further differentiation into sub-modules is made by distinguishing between sea ice or open water, leading to five modules in total. One of the main differences between the new and the original AMSU-B algorithm is the use of the emissivity relationship in equation 19 for applying the extended-TWV retrieval
over open water areas. This allows for retrieval over a greater spatial domain as the original algorithm could only use the mid-TWV sub-algorithm over open water for TWV values up to $7\,kg/m^2$.

The other addition over the original AMSU-B algorithm is that the mid-TWV sub-algorithm differentiates between sea ice/land and open water and has different retrieval equations for each
of the two cases. We believe that this is a more physically consistent treatment than using the equal emissivity assumption. The specific open water module uses the regression relationship in equation 24, while the sea ice/land module uses the equal emissivity assumption. In the

original algorithm mid-TWV retrieval over open water also operates under the equal emissivity assumption.

The algorithm for low-TWV uses AMSU-B channels 20, 19, and 18 for the retrieval range 0 to $1.5\,kg/m^2$. These are the band channels around the strong water vapour line at 183.31 GHz, and have the best accuracy and present the lowest error as the assumption of equal surface emissivity is valid for these three frequencies.

The mid-TWV algorithm using AMSU-B channels 17, 20, and 19 takes over retrieval up to $7\,kg/m^2$. It is assumed to be independent of the surface type but the retrieval error might increase when approaching the upper retrieval limit. The assumption of equal emissivity is still used over sea ice covered surfaces, even though there are some differences because of the introduction of the 150 GHz channel instead of the $183\pm1$GHz channel. Because of this difference in real surface emissivity a positive bias of up to $0.5\,kg/m^2$ is possible (Selbach, 2003). Over areas with sea ice concentration below 80% SIC the specific open water sub-module of the mid-TWV algorithm uses the ratio of reflectivities at 183 and 150 GHz in order to account for the different surface emissivities of open water at these frequencies.

The extended-TWV module uses the channels 20, 17, and 16 to retrieve TWV in the range 7 - 15 $kg/m^2$. Previously, the retrieval from these channels was restricted to sea ice regions and because of the simplified treatment of the surface emissivity difference, the error can reach $3\,kg/m^2$. Similarly with the mid-TWV module above, a dedicated open water version of the extended-TWV module uses the ratio of reflectivities at 89 and 150 GHz over scenes with mixed water and sea ice (the).

Due to of the specific channel triplet used by each sub-module, the set of four calibration parameters has to be determined for the low TWV, mid-TWV and extended-TWV cases, and two extra sets for the new mid and extended-TWV retrieval scenarios over open water.

## 2.8 How the retrieval works

One of the critical points in all of the AMSU-B algorithms is to correctly identify when one certain triplet of channels becomes saturated in order to switch to the next available triplet. In

the initial Antarctic algorithm paper by (Miao et al., 2001), this was accomplished by checking the sign of the brightness temperature difference using the condition in Eq. (11).

In order to extend the coverage while keeping the retrieval error reasonably low, the constraint above was relaxed in (Melsheimer and Heygster, 2008) by allowing the brightness temperature difference to go slightly above zero so that in the end, the following condition is applied:

$$T_{b,j} - T_{b,i} < F_{i,j}. \tag{26}$$

$F_{i,j}$ is the focal point calculated for a particular channel triplet. It usually has a value of a few Kelvin. The retrieval will work as long as the sign of the brightness temperature differences ratio $\eta_c$ is positive.

This relaxed condition allows the channel triplet to be used until its high absorption channel approaches saturation and allows an extension of the retrieval range of that triplet by up to $1\,\mathrm{kg/m^2}$. The disadvantage of this relaxed condition is that the retrieval error also increases when a channel in the triplet is close to saturation.

By mapping (not shown) the pixels according to the conditions used in their retrieval with the original algorithm we found that the values near the saturation limit retrieved under the relaxed conditions in most cases account for open water or mixed water/sea ice surfaces. This is where the equal emissivity assumption breaks down because the microwave emissivity of water is much lower than that of sea ice producing an increased retrieval error. In this new algorithm we propose to use a specific method for those areas.

For the mid and extended TWV range algorithms there is a further differentiation in the modules used between sea ice and open water surfaces. Based on our experience, a threshold of 80% sea ice concentration was chosen in order to differentiate the typically dry areas of high sea ice concentration in the central Arctic and the regions with a larger ratio of open water to sea ice where higher atmospheric water vapour loads are expected. In these peripheric regions the new algorithm is employed. In all areas with sea ice concentration above 80% the retrieval technique from Melsheimer and Heygster (2008) is used, which is better suited for the very low atmospheric water vapour values encountered in this region.

To illustrate how the new algorithm works with these new sets of conditions we will present each step, with its differences to the previous method.

1) The algorithm begins by using the full set of five brightness temperatures of the AMSU-B instrument. In the previous method, it would first identify pixels where the conditions

$$T_{b,19} - T_{b,18} < F^L_{19,18} \text{ and } T_{b,20} - T_{b,19} < F^L_{20,19}$$

hold true. Here $F^L_{19,18}$ and $F^L_{20,19}$ are the pairwise focal points for channel pairs (18,19) and (19,20). This condition fulfilled allows for the channel triplet (18,19,20) to be used for the range up to $2\,\text{kg/m}^2$. Because the retrieval range of the first two channel triplets (low and mid range) overlaps around $2\,\text{kg/m}^2$ we will keep the stricter condition from the Antarctic algorithm (Miao et al., 2001).

$$T_{b,19} - T_{b,18} < 0, \ T_{b,20} - T_{b,19} < 0.$$

For these pixels the low-TWV algorithm is applied.

2) If the first condition fails, the second one is checked. In the previous method this was

$$T_{b,20} - T_{b,19} < F^M_{20,19} \text{ and } T_{b,17} - T_{b,20} < F^M_{17,20}.$$

Continuing from the strict zero threshold condition for the low-TWV, the new condition threshold is

2a) $T_{b,19} - T_{b,18} \geq 0$ and $T_{b,20} - T_{b,19} < 0$ and $T_{b,17} - T_{b,20} < 0$.

This test is performed for pixels with over 80% sea ice concentration. Where this is true, the original mid-TWV retrieval is used.

2b) Over open water and scenes with ice concentration below 80% in the mid TWV range,

the algorithm now uses a somewhat different condition

$$T_{b,19} - T_{b,18} \geq 0 \text{ and } T_{b,20} - T_{b,19} < F_{20,19}^X \text{ and } T_{b,17} - T_{b,20} < F_{17,20}^X$$

Condition 2b) means that the pixels which were previously retrieved under the equal emissivity assumption over open water will now be treated separately according to their surface type taking into account the surface emissivity component. Those pixels that are at the saturation limit for the mid range, but do not contain open water are being flagged for further processing with the extended range sea ice algorithm. This would include pixels retrieved above land in less dry conditions (in the Arctic case this means TWV $> 2 \text{ kg/m}^2$).

3) For applying the extended-TWV algorithm, the remaining pixels are tested for

$$T_{b,17} - T_{b,20} < F_{17,20}^X \text{ and } T_{b,16} - T_{b,17} < F_{16,17}^X,$$

and where true, processed. In addition to this test for channel saturation, the data is again classified for its surface type, and only sea ice or open ocean areas are kept excluding land. This surface classification is done for all channels by a comparison with sea ice concentration maps derived using the ARTIST Sea Ice concentration retrieval (Spreen et al., 2008) from SSMIS (Special Sensor Microwave Imager / Sounder) or AMSR-E (Advanced Microwave Scanning Radiometer - Earth Observing System) data depending on the retrieval date.

## 2.9 Comparing the new retrieval with other TWV retrieval products

For a comparison we use daily averages for thirty consecutive days in each of four months, September, March, June and December which represent the variability of the atmospheric parameters and sea ice extent. September and March represent the two extremes of sea ice extent. The minimum extent in September is usually coupled to warmer air and higher atmospheric water vapour load. The maximum extent in March corresponds to lower air temperatures and a drier atmosphere. June and December represent transition periods between the two extremes.

In order to obtain a bigger data sample we ran this analysis using daily averaged data for three consecutive years from 2007 to 2009. The geographical domain we chose represents the entire Northern Hemisphere above 50° N latitude. Though the very first retrieval was targeted towards the Central Arctic region, this was because the atmospheric water vapor load over this region was low enough for the algorithm to retrieve. After the subsequent extensions of the retrievable TWV range, the geographical domain for applying the algorithm has increased as well. For the purpose of this work we have arbitrarily chosen this 50° N limit because that is the approximate latitude where the water vapor load in winter is low enough to allow for a time consistent retrieval with the newest version of the algorithm. All landmasses in this region are also included because the method is able to retrieve TWV there if the values are low enough. For example Greenland is always included in the retrieval because the atmosphere above it is dry throughout the year.

From AMSU-B data we produce two versions of the TWV product, one retrieved with the original Melsheimer and Heygster (2008) method and the other with the new algorithm. The calibration parameters we derived separately for each channel combination with the corresponding linear relationships between surface emissivities from the same batch of radiosonde TWV data.

First we want to see how the new retrieval method performs against the original one and hence test both methods against two other TWV products chosen as benchmarks in this field. The first benchmark is the ECMWF (European Center for Medium Range Weather Forecasts) ERA-Interim (Dee et al., 2011) reanalysis model data from which TWV values were derived.

The second dataset is the TWV product from Remote Sensing Systems (RSS) that uses AMSR-E brightness temperatures and an algorithm adapted from (Wentz, 1997). This retrieval algorithm has been developed for global coverage and works over all ice free ocean surfaces. Because of this it can cover a large range of TWV values (0 - 75 kg/m$^2$) but it was not specifically tuned for the dry Arctic conditions. This data set covers the entire nine year lifespan of AMSR-E and has been used for creating derived products (Smith et al., 2013) and validated against ship based observations (Szczodrak et al., 2006) and hence is considered a good benchmark against which the new AMSU-B retrieval can be compared.

A third test dataset was obtained from the (Bobylev et al., 2010) algorithm. This method is a neural network based approach designed specifically for the ice free regions in the Arctic. As a training dataset for the neural network the authors used radiosonde data from Russian polar stations. The method is able to retrieve low TWV values over open ocean areas using the same AMSR-E instrument as the RSS TWV product with similar TWV value ranges. This neural network approach is proven to have a smaller root mean square error than the Wentz global algorithm used in the RSS TWV product. These three retrievals are compared over one common valid spatial domain (only open water) while using the ECMWF TWV data as a reference point.

## 3 Results and Discussion

### 3.1 Comparison of results to Melsheimer and Heygster (2008)

Independent of the comparison benchmark, an important difference between the original and the new retrieval is the area the algorithm can cover for retrieving TWV in the Arctic. Because both algorithms use the same instrumental input, a one to one comparison of coverage represented as the number of valid retrievable pixels is possible.

Figure 4 shows two examples in the coverage difference between the original and the new retrieval, for one summer and one winter day. The largest benefit of the new retrieval is that large areas in the North Atlantic and Pacific oceans can now be covered. The only limitation of the method is the amount of water vapor present in the atmosphere and, for the extended range module, the presence of either an open water or sea ice covered surface. In both the summer and winter case the new method has a larger coverage area, with the biggest difference being seen in the summer case. To add to this analysis, Figure 5 presents the frequency of retrieval for the new and the original retrieval versions when looking at the months of June and December for the whole three year interval studied. Each pixel value represents the number of times that particular region has been present in the daily retrieval maps for the test time series. The minimum value shown is five while the maximum is 90 days. As in the one day example of Figure 4, the increase in coverage for the month of June is evident, with the addition of North

Atlantic and Pacific Ocean regions where the water vapor values are within the retrieval range. For December the frequency of retrieval has been improved where these same ocean regions can now be retrieved more consistently with the new algorithm.

When comparing the two AMSU-B retrieval methods for the whole testing time series we look at monthly averages compiled from swath data for each method. The comparison was done for the representative four months of each year from 2007 to 2009 in order to see how the total area of retrievable pixels is affected by the new method (Fig. 6). In the colder months of March and December the benefit of the new method is marginal because of the larger sea ice extent (when compared to the summer months) and the overall low water vapour burden of the atmosphere. In these months we can observe a small increase of 17.4% and 21.18% respectively, compared to the coverage of the original algorithm. For September and June the number of retrievable pixels increase by 152% and 176%, respectively of the original number (Fig. 6). This change is significant considering that these areas were beyond the retrieval capabilities of the original method.

ECMWF ERA-Interim reanalysis data was used as reference in order to compare the original method and the new one. The ECMWF TWV information was directly compared to collocated daily averages from both algorithms. In terms of correlation with the ECMWF, the two algorithms vary significantly (Fig. 7).

The new method matches the correlation of the original one for the month of March (0.86), and even surpasses it for December (0.82 vs 0.77). In the months with moist conditions and lower sea ice extent, June and September, the correlation drops to 0.36 and 0.32 vs 0.57 and 0.61, respectively. Comparing this with the results in Fig. 6 shows that in the months where the spatial contribution of the improved algorithm is greater, the correlation drop is more significant. For a more detailed look into the differences between the original and new AMSU-B methods a side by side comparison is presented in Figure 8. Here the original and new algorithm are compared with each other and then individually with ECMWF TWV over the same common domain valid for both retrievals. One major difference between the two algorithms is in the way mid-TWV retrieval is performed. The new algorithm uses a dedicated open water sub-module while the original algorithm treats all surface types the same in this retrieval range. While the

small differences in calibration parameters can cause minor differences in the low-TWV domain where the retrieval equations are identical, the different treatment of the surface type causes a larger deviation between the results at the upper limit of the mid-TWV range. Another modification that has an impact on the new retrieval are the differences in trigger thresholds which cause the algorithm to switch to a different retrieval module. These thresholds and the differences between the new and original versions have been described in Section 2.9. Because of the stricter switching condition, the new algorithm switches to the extended range retrieval module earlier than the original algorithm and retrieves higher TWV values. These data points which are retrieved with the extended range module in the new version and with the mid-TWV module in the original one can be seen as a plume of higher AMSU-B new TWV values which deviate from the identity line in the left side plot of Figure 8. The comparison of the new algorithm with ECMWF TWV in the right most plot of the same figure indeed shows a similar cloud of overestimated data points closer to the maximum retrieval limit of the algorithm.

From Figure 9 where the new algorithm is compared against ECMWF TWV over its full spatial domain of valid retrievals the higher uncertainty of data points with larger TWV value is evident. The scatter in the retrieved data increases with the retrieved TWV value. When the retrieved values approach $15\,\text{kg/m}^2$ the involved channels are near the saturation limit so that the true value may well exceed this retrieval limit. As a practice for future studies we recommend to only use retrieval values up to $14\,\text{kg/m}^2$. It is however important to note that the majority of data points retrieved fall within the $0\text{-}6\,\text{kg/m}^2$ interval which matches well with the model data. The extended range retrieval represents the maximum coverage that can be obtained with this instrument and algorithm combination. This increased coverage with the price of increased uncertainty represents the only way to consistently cover regions where a complete data gap existed before in mixed sea ice/open water regions. In these areas other retrieval methods, like the AMSR-E based ones presented in Section 2.9, cannot function because of the presence of sea ice while the original AMSU-B algorithm could not retrieve anything because of the presence of open water and high TWV values.

When considering the difference between the ECMWF data and the AMSU-B retrieval, the highest bias is again seen in the warmer months (Fig. 10). Following the correlation results

shown in Figure 7, the higher variability of the new AMSU-B retrieval is confirmed by the root mean square difference against ECMWF data which are represented by the error bars in Figure 10. The bias behaviour versus ECMWF has changed from the original to the new algorithm. Both retrievals follow the same pattern of low bias in winter and higher bias in summer months.

This can be explained by seasonal variability in the mean water vapor load of the atmosphere. In winter months the mean ECMWF TWV is usually below 8 kg/m$^2$ which represents the lower range of possible retrieval for the mid-TWV module. In summer months this average value is between 13 and 15 kg/m$^2$ which sits in the upper range of the extended TWV module, the module most susceptible to higher uncertainties. The change in winter months is an improvement

over the original, with the bias versus ECMWF decreasing throughout the winter months. For the winter months the new method registers an increase in bias when compared to the original in the month of June for 2007 and 2008, while scoring a lower value in 2009. These differences are small however and we prefer to focus on the seasonal variability which is matched between the two algorithm versions with one notable exception. While similar in absolute value, the new

method bias in the month of September for each of the three years shows a sign inversion when compared to the original method. When the latter was on average underestimating TWV against ECMWF data, the new algorithm shows an overestimation for the same months. Considering the technical differences between the two algorithms, this change in behaviour can be attributed to the split of the extended algorithm in sea ice and open water sub-modules respectively. The

open water extended range module of the new algorithm returns overestimated TWV values when compared to collocated ECMWF data. This confirms the behaviour observed in Figure 8 where when compared over the same domain, the new algorithm extended TWV module seems to overestimate the retrieval when compared to model data. Another agreement on the behaviour of the new algorithm retrieval can be seen between Figures 8 and 10 where the large variability

of the new algorithm retrieval at higher TWV values is evident.

## 3.2 Intercomparison of New AMSU-B retrieval, RSS TWV and Neural Network method

With this comparison we explore how well the new AMSU-B retrieval matches other retrieval methods like the RSS TWV product and the retrieval based on the neural network approach. As

a benchmark data set we chose the ECMWF TWV data. Even though circulation model data in general does not represent a source for validating satellite retrieval products, it may serve as a consistent data set from which the retrievals should not deviate too much. While a match between the model data and any one retrieval does not represent validation, any large deviations from the model could signify errors in the retrieval data. We expect to conclude whether or not the output of the new AMSU-B algorithm is reasonable and warrants further development and validation efforts.

To judge if a method-specific bias exists versus the model and if a seasonal variability is present, monthly differences were calculated between the ECMWF data and each retrieval using daily averages. To allow an intercomparison between them all retrieval products plus the ECMWF model data are gridded to one common grid so that the spatial domain used in the comparison always represents the largest common domain for all represented data sources. In Figure 11 the individual monthly bias values are shown for each of the three methods over open ocean areas above $50°$ N. The method specific bias as well as the ECMWF mean value was calculated for the same domain common to all methods.

Two of the retrievals compared here show a seasonal variation in bias. Both the AMSR-E based neural network and the AMSU-B retrieval underestimate TWV when compared to the model with the former showing the lowest bias throughout the entire dataset. The AMSR-E RSS retrieval presents a small constant overestimation of around $1\,kg/m^2$ throughout the entire dataset without any strong seasonal characteristics. This behaviour of the RSS product matches the findings of (Bobylev et al., 2010) where it is shown that the global Wentz retrieval tends to overestimate water vapour content in the dry conditions of the Arctic. The AMSU-B retrieval shows good results in winter when its low bias values place it close to the neural network retrieval with performance comparable to the RSS retrieval. The AMSU-B method shows much higher negative bias values in summer. Then, the average TWV values for the ice-free ocean frequently surpass the saturation value of $15\,kg/m^2$, so that the AMSU-B retrieval works at the limits of the algorithm where the higher uncertainty is assumed.

Figure 12 top displays the average TWV value of each retrieval method calculated over the same domain as in Figure 11. The results of the neural network method closely follow the model

value throughout the year. The new AMSU-B retrieval matches the model almost as well in the winter months while in the summer months it present a more pronounced underestimation with respect to the ECMWF TWV average values.

Because the intercomparison with the two AMSR-E based methods can only be done for ice-free ocean areas, the assessment of the new AMSU-B retrieval cannot be complete without taking into account the strong points of this approach, which is that it can retrieve TWV over the entire Arctic scene including land and sea ice-covered as well as open ocean surfaces. To this end a final comparison is made for average TWV over the whole valid domain of the new AMSU-B retrieval (Fig. 12 bottom). The performance increase throughout the year is obvious when compared to the open ocean regions alone. The average retrieved TWV for winter months matches better with the model while the overestimation for summer months, although still present is greatly diminished. Also important to note is how much the average ECMWF TWV value decreases in the summer months (from $15 \, \text{kg/m}^2$ to $10.5 \, \text{kg/m}^2$ in June and from $12.8 \, \text{kg/m}^2$ to $11.9 \, \text{kg/m}^2$ in September) once the ice-covered regions of the Central Arctic are added to the comparison. This shows the difference between the average atmospheric water vapour load for the entire Arctic when including all sea and land ice covered regions and the higher TWV ice-free Arctic Ocean.

## 4 Conclusions

Based on the previous work of (Miao et al., 2001) and Melsheimer and Heygster (2008), we present a method to achieve a more complete coverage for TWV retrieval in the Arctic region. The previous method was able to retrieve TWV over all surface types for atmospheric water vapour loads up to $7 \, \text{kg/m}^2$ and over sea ice for up to $14 \, \text{kg/m}^2$. The new method extends the coverage of the maximum range retrieval over open water where higher values for TWV are frequently found, especially in summer when the sea ice extent is small. This aspect has become even more important in the last decade with the dramatically decaying Arctic summer sea ice.

Because of the unique way in which each of the three cases of channel coupling and surface types are handled, the algorithm has become more complex. Each of the five sub-algorithms

is designed with a set of specifically derived calibration parameters and three of them (Mid TWV range-open water, extended TWV range, extended TWV range-open water) use retrieval equations that take into account a linear regression between the surface emissivity at 89 and 150 GHz.

The modifications brought to the AMSU-B retrieval are meant to improve the retrieval over mixed areas of sea ice and open water by including a comprehensive treatment of the open ocean emissivity. Another improvement is the increase in retrievable area under the condition of a recommended upper limit of $14 \, \text{kg/m}^2$ for valid values.

The new method shows an improvement both in correlation (Fig. 7) and bias (Fig. 10) with ECMWF data for the winter months, and a large increase in coverage for the summer months (Figs. 4, 5, 12) because of the dedicated treatment of open water emissivity in the mid and high TWV ranges. When compared to ECMWF reanalysis data, the new algorithm is shown in Fig. 10 to have a higher RMSD than the original one, with average ranges from $1.86 \, \text{kg/m}^2$ ($1.08 \, \text{kg/m}^2$ previously) in March, up to $5.67 \, \text{kg/m}^2$ ($3.79 \, \text{kg/m}^2$ for the original algorithm) in September. This difference can be explained by the additional area covered with the new algorithm which adds high TWV value and high uncertainty data points to the comparison. This can also be seen from the 2D histograms in Figures 8 and 9. For the month of September we have an increase in average coverage of 174% as compared to 17,4% for March (Fig. 6). This accounts for all of the open ocean areas where the extended range sub-algorithm can now be employed with the connected higher error margins previously acknowledged in Melsheimer and Heygster (2008).

When comparing the new method with two established algorithms for retrieving atmospheric water vapour over open ocean it is shown that the new AMSU-B retrieval method has similar performance in winter months (Figs. 10, 11). As these two AMSR-E based methods are restricted to open water areas where the atmospheric water vapour load is higher in the summer months, the AMSU-B algorithm performance decreases correspondingly in these conditions because of the relatively low saturation limit of $15 \, \text{kg/m}^2$. The neural network approach by (Bobylev et al., 2010) ranks first as the most accurate retrieval when compared to ECMWF model data over open water only. The new AMSU-B method scores similarly in winter months

while the RSS TWV product based on Wentz (1997) which was calibrated for global operation displays a low but constant positive bias throughout the seasonal cycle. It is important to note however that the strength of new AMSU-B method is that it can seamlessly retrieve atmospheric water vapor loads over a large spatial domain which includes land and sea ice besides the open ocean areas where the established retrieval products have already proven themselves. The accuracy of the new AMSU-B retrieval relative to ECMWF data increases when the entire Arctic region is taken into account, including all sea and land ice areas. This demonstrates the capabilities of the method to retrieve TWV simultaneously over all surface types in the specific atmospheric conditions of the Arctic and adjacent regions. The new algorithm extends the spatial coverage in the warmer months where higher TWV values can be retrieved over open water and mixed surface regions. Data gaps present in the original method results are covered for the winter months as well. These high TWV measurements however are also connected with larger uncertainties as the algorithm is working near the limits of the instrument sensitivity.This approach requires a trade-off between achieving a high spatial coverage of the polar region, while accepting the lower accuracy dictated by instrument limitations, or using multiple instruments/methods each with their inherent collocation and accuracy issues to cover the same region.

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

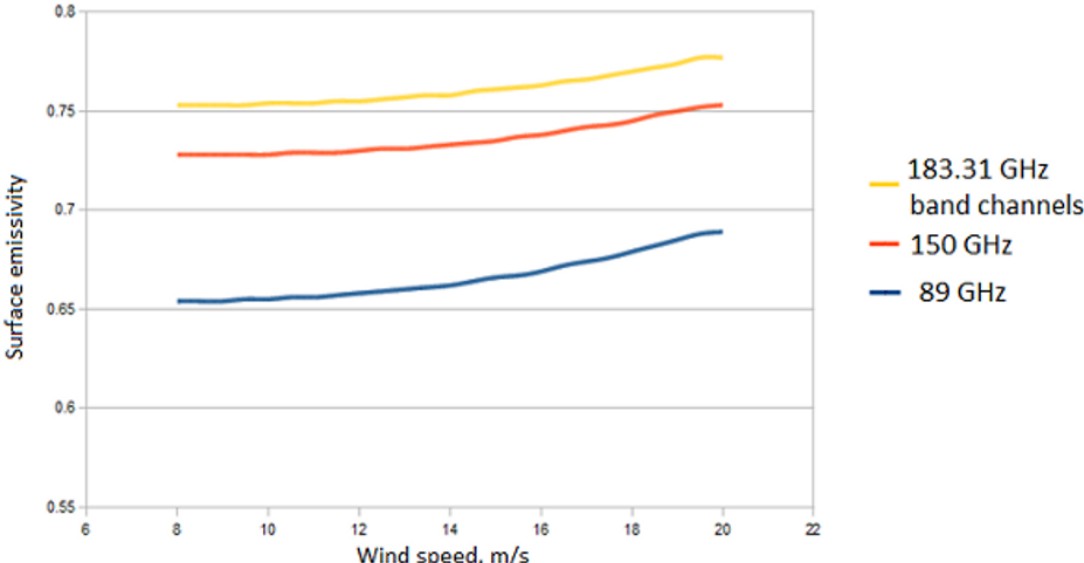

**Fig. 1.** Ocean surface emissivities' dependence on wind speed.

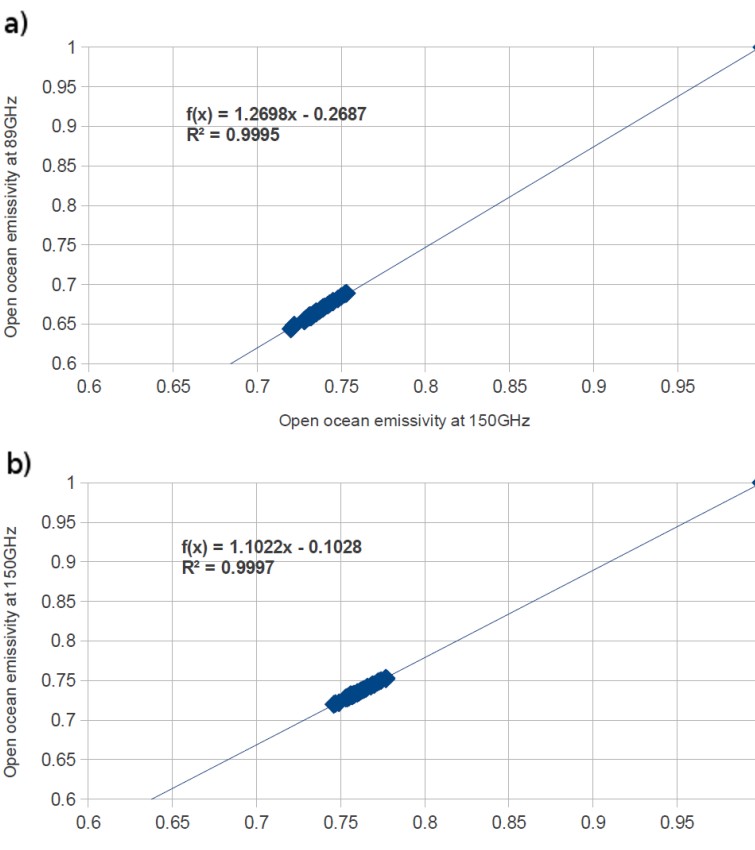

**Fig. 2.** Regression plot for ocean surface emissivity at 89 and 150 GHz (panel a) and at 150 and 183 GHz (panel b).

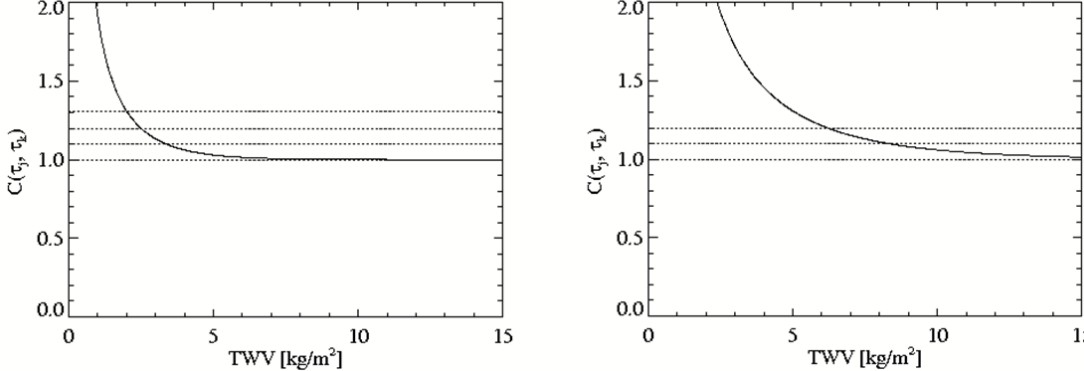

**Fig. 3.** $C(\tau_j, \tau_k))$ parameter for mid-TWV algorithm (left) and for extended-TWV algorithm (right). The dashed horizontal lines represent the variability interval for the $C(\tau_j, \tau_k))$ parameter inside the TWV range corresponding to each case.

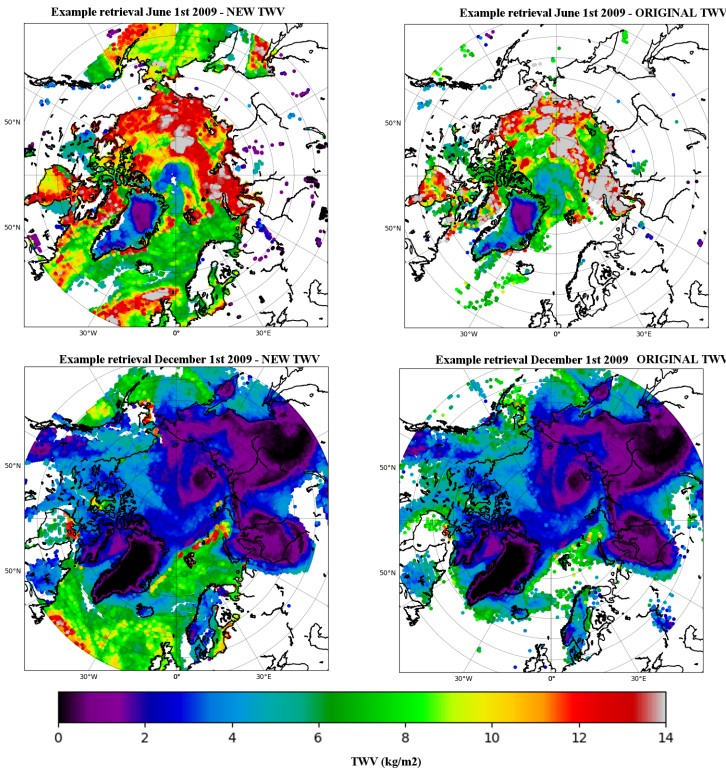

**Fig. 4.** Daily TWV maps of the Northern Hemisphere obtained from the new algorithm (left column), compared to the original AMSU-B algorithm (right column). The days represented here are 1st of June 2009 for the top row, and 1st of December 2009 for the bottom row.

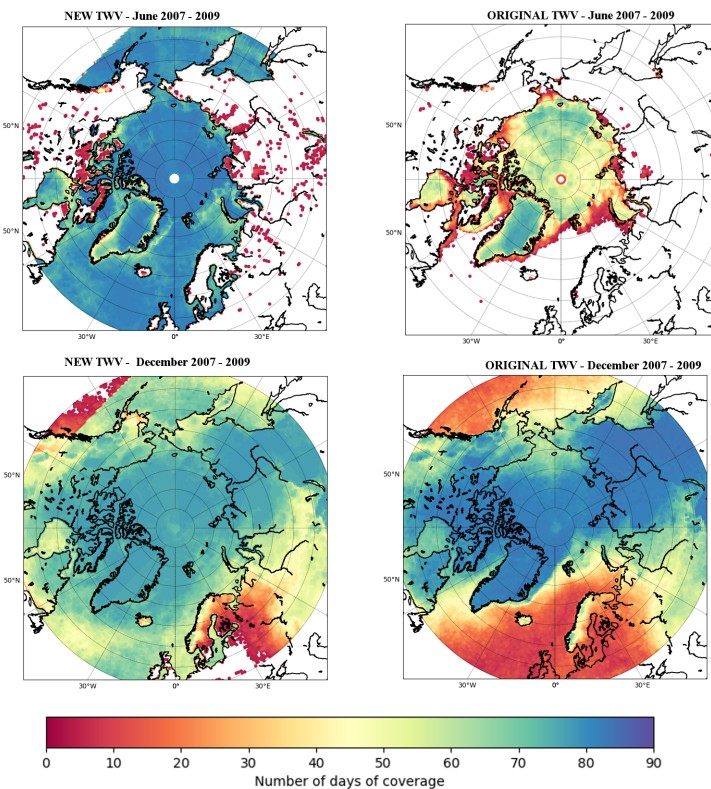

**Fig. 5.** Number of days of coverage for the Northern Hemisphere for one typical summer and one winter month, over three years. The top row represents the month of June for the 2007-2009 interval, while the bottom row is December for the same three years. The left side column shows results for the new algorithm, while the right side for the original AMSU-B algorithm.

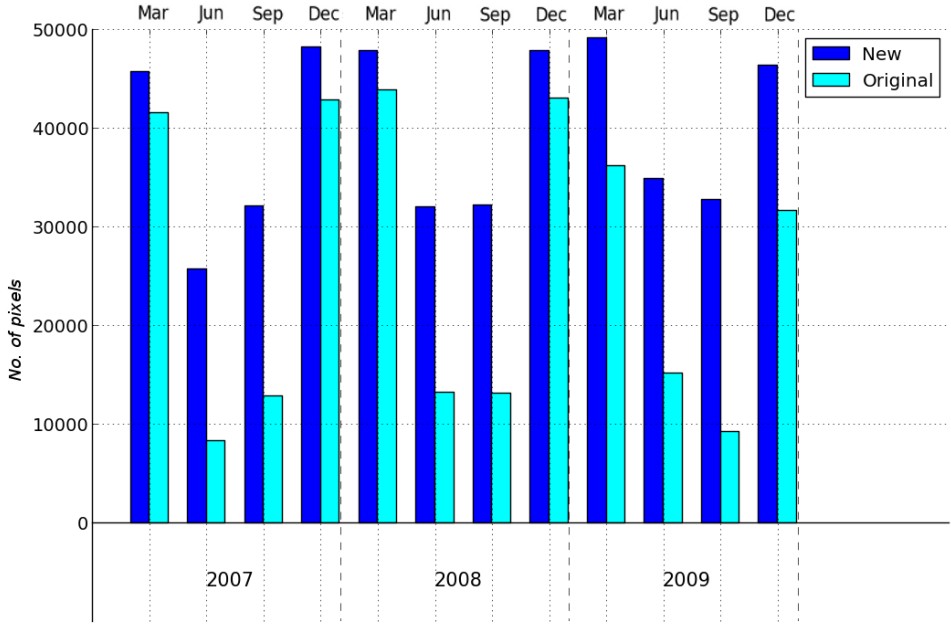

**Fig. 6.** Monthly average number of pixels covered by each method for the test interval of three years.

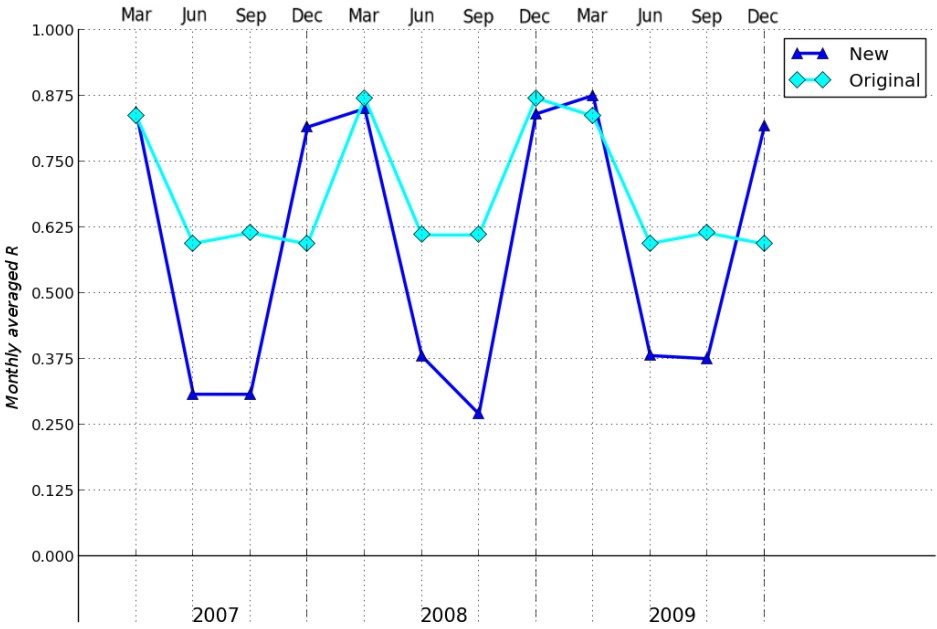

**Fig. 7.** Correlation for the original and the new AMSU-B TWV retrieval with ECMWF data.

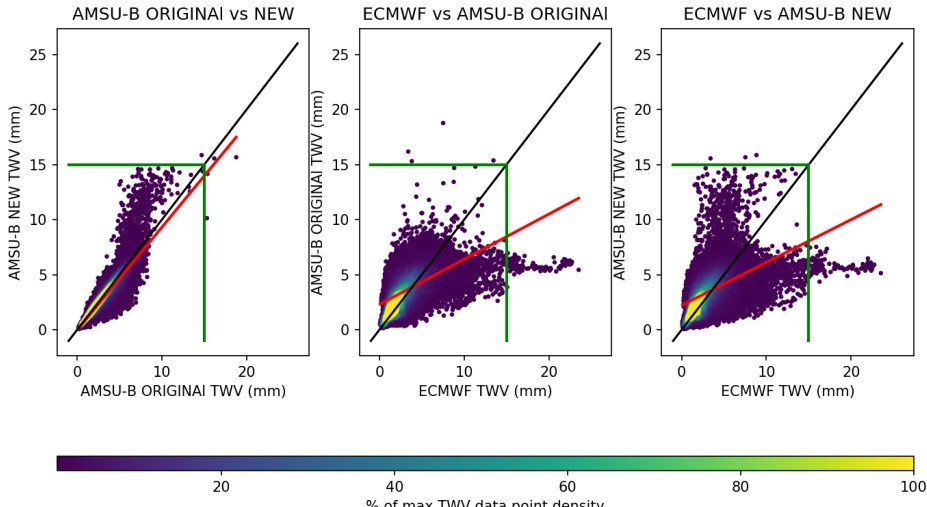

**Fig. 8.** Scatter plots of the original and new AMSU-B TWV retrievals vs ECMWF TWV. The spatial domain is the common valid domain of both methods. The entire test dataset of 12 months over three years is represented. The black line is the identity line, while the red line represents the data linear regression and the two green lines show the $15(kg/m^2)$ saturation limit of the extended range TWV retrieval module.

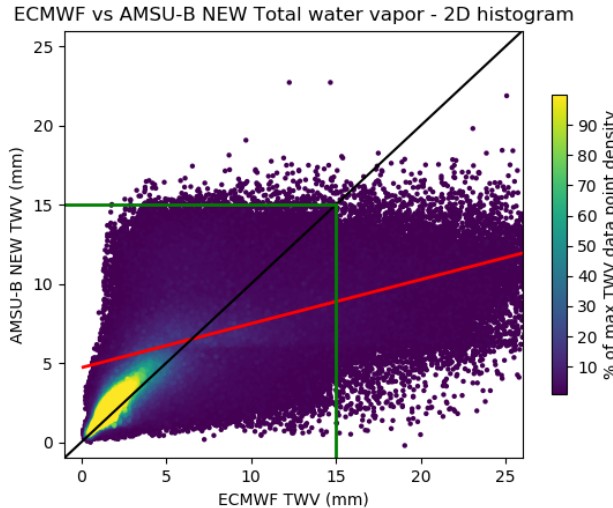

**Fig. 9.** New AMSU-B TWV retrieval vs ECMWF TWV. The spatial domain is the full valid domain of the new algorithm. The entire test dataset of 12 months over three years is represented. The black line is the identity line, while the red line represents the data linear regression and the two green lines show the $15(kg/m^2)$ saturation limit of the extended range TWV retrieval module.

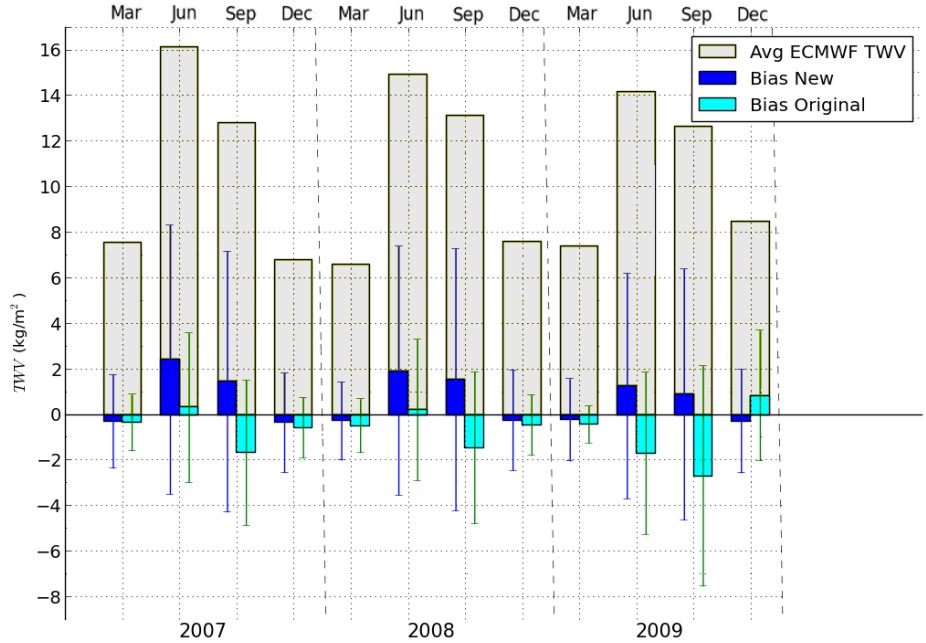

**Fig. 10.** Bias comparison of the Original and New AMSU-B TWV retrieval versus ECMWF data. Error bars represent the RMSD between the retrieved TWV data and ECMWF TWV.

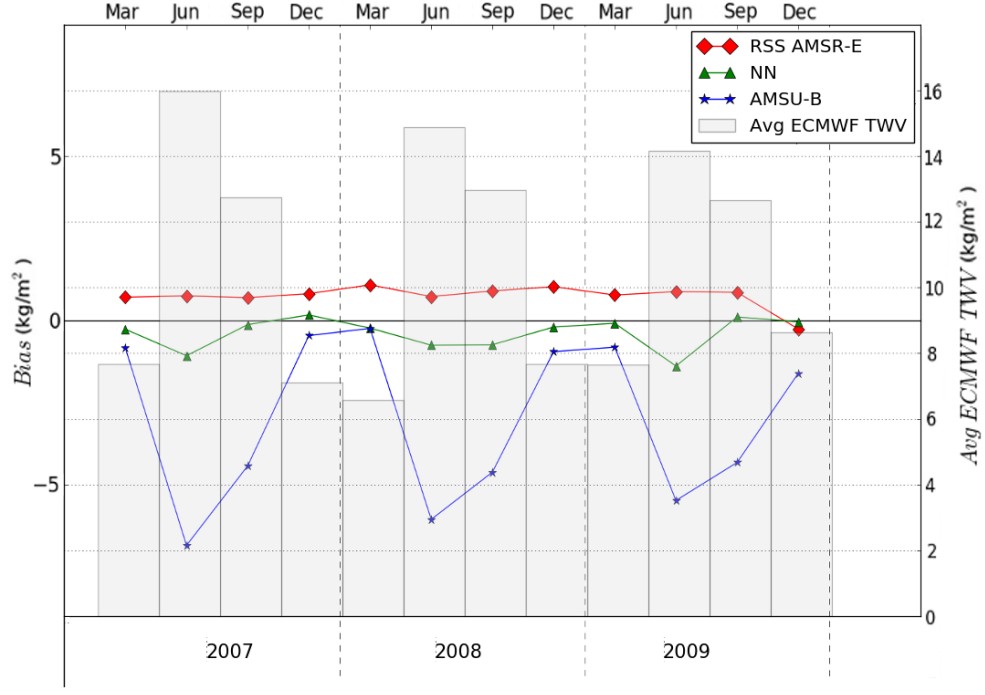

**Fig. 11.** Bias for the New AMSU-B, AMSR-E RSS and AMSR-E Neural Network retrievals over open ocean versus ECMWF average TWV value. The curves represent the monthly averaged bias values for each of the three algorithms with the scale on the left side. The vertical columns in the background represent the monthly mean TWV value from ECMWF data with the scale on the right side.

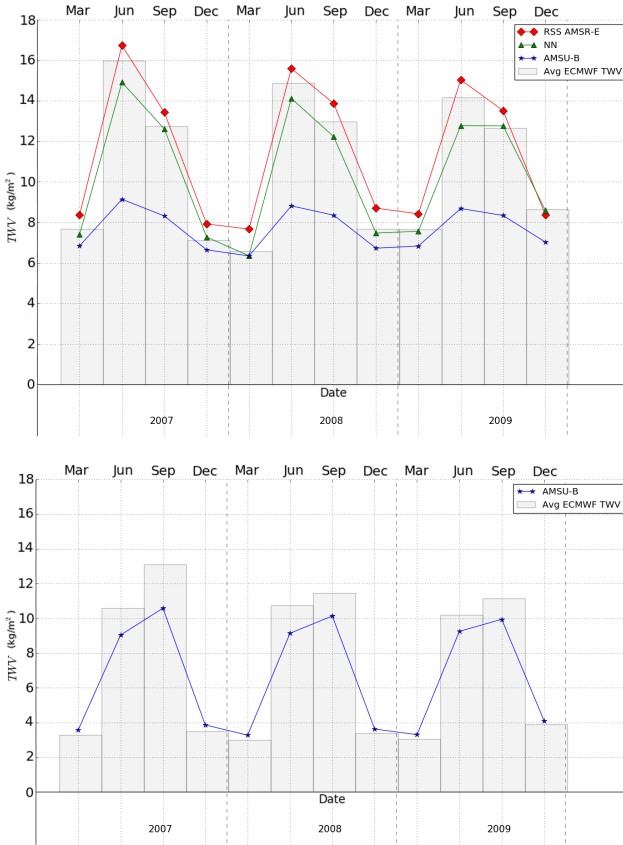

**Fig. 12.** Average TWV value for the three tested retrievals plotted over ECMWF average TWV value for open ocean areas (top) and New AMSU-B alone plotted over ECMWF average TWV value including all open water, land and sea ice-covered regions of the Arctic where valid values can be retrieved (bottom).The curves represent the monthly averaged TWV values for each algorithm. The vertical columns in the background represent the monthly mean TWV value from ECMWF data.

**Table 1.** Comparative structure of three TWV retrieval algorithms. SI represents sea ice only and OW - open water as surface types where the individual modules can be applied. L,M and E represent low, mid and respectively extended range TWV retrieval modules. *The Miao algorithm was developed for the SSM/T2 instrument and for the Antarctic region.

| Method | Sub-modules | Channel freq. (GHz) | Channel no. | TWV (kg/m$^2$) | Surface |
|--------|-------------|---------------------|-------------|----------------|---------|
| Miao algorithm | L-TWV | 183.31±1,±3,±7 | 2,3,4* | 0 – 1.5 | All |
| | M-TWV | 183.31±3,±7,150 | 3,4,5* | 1.5 – 6 | All |
| Original AMSU-B | L-TWV | 183.31±1,±3,±7 | 18,19,20 | 0 – 1.5 | All |
| | M-TWV | 183.31±3,±7,150 | 19,20,17 | 1.5 – 7 | All |
| | E-TWV - SI | 183.31±7,150,89 | 20,17,16 | 7 – 15 | SI |
| New AMSU-B | L-TWV | 183.31±1,±3,±7 | 18,19,20 | 0 – 1.5 | All |
| | M-TWV | 183.31±3,±7,150 | 19,20,17 | 1.5 – 7 | SI/land |
| | M-TWV - OW | 183.31±3,±7,150 | 19,20,17 | 1.5 – 7 | OW |
| | E-TWV - SI | 183.31±7,150,89 | 20,17,16 | 7 – 15 | SI |
| | E-TWV - OW | 183.31±7,150,89 | 20,17,16 | 7 – 15 | OW |