# Peer review of "Retrieval of Total Water Vapour in the Arctic Using Microwave Humidity Sounders"

_Atmospheric Measurement Techniques, 2017_

## Referee Comment (RC1) · Anonymous Referee #1 · 26 Nov 2017

The manuscript by Scarlat et al. discusses an attempt to improve upon the total water vapor (TWV) retrieval described in Melsheimer and Heygster (2008). They extend the coverage of the retrieval to low ice cover and ice-free areas in the Arctic by using modeled emissivity values. They find that their new algorithm is able to retrieve TWV over more of the Arctic than the previous algorithm but with larger errors in the newly measured regions. This topic is of importance because in situ measurements of water vapor in the Arctic are sparse and satellite retrievals are still limited in a number of ways, which this paper attempts to address. The manuscript is well written and I recommend publication in AMT after considering my comments below.

General comments:

1) A table of all the algorithms would be immensely helpful. Include:
    - The channels used
    - The corresponding AMSU-B channel numbers
    - The expected TWV range measured
    - The surface types allowed
You may want to also include the original Miao and Melsheimer and Heygster algorithms for comparison.

2) It would be helpful to have a plot of new AMSU-B TWV vs. ECMWF as a function of TWV to show how the errors get worse as you approach the saturation limit of 15 kg/m$^2$.

3) Many of the equations presented in this manuscript are directly in Melsheimer and Heygster (2008) and it may not be necessary to duplicate them here.

4) The authors refer to an "original" algorithm multiple times throughout the manuscript. It should be clarified that this is referring specifically to the Melsheimer and Heygster (2008) algorithm.

Specific comments:

Abstract:
- You don't mention what you found. How did the new algorithm perform and what are you general conclusions? This needs to be in the abstract.

Introduction:

- "*However this AMSU-B based method...*"
    Miao et al. (2001) uses SSM/T2. Regardless, please define acronyms the first time they are used.

- "*But the emerging errors were deemed acceptable as a trade-off for extending the retrieval range from 1.5-2 kg/m² (for only the three band channels) up to 7 kg/m² (for two band channels together with the 150 GHz channel).*"
  The use of "band channels" to refer to the 183 GHz channel specifically is confusing.

- "*Melsheimer and Heygster (2008) extends the TWV retrieval range over sea ice by including the 89 GHz channel into the retrieval.*"
  Include a short sentence on why including the 89 GHz channel is physically useful (it doesn't saturate as quickly).

- Much of the paragraph starting at P4 L9 feels like it belongs in section 2.2. Too many technical details for the introduction.

- It may be useful to include a statement at the end of the introduction explaining what will be described in remaining sections (e.g. "Section 2.1 provides a discuss of the RT...").

Methods

2.1

- Need to make it clear early in the paper that you're primarily analyzing AMSU-B measurements in this study.

- "*A down-looking microwave radiometer*"
  You may want to note that this is the same type of instrument as a "humidity sounder".

2.2

- "*In the original paper (Miao, 1998)...*"
  If this is the original paper that this work is based on, it deserves discussion in the introduction.

- "*Because the Ts term is the same for both brightness temperatures, it has disappeared from Eq. (2) as a result of the subtraction.*"
  Technically $T_s$ is still in the $b_{ij}$ term and thus in Eq. (2).

- "*To find the relationship between the measured brightness temperature and the water vapour absorption we require the third brightness temperature measured in channel k.*"
  A brief physical explanation of why three channels are necessary would be appreciated for people unfamiliar with microwave TWV retrievals.

- *"Compared to the first two terms under the exponent, the quadratic term can be neglected…"*
  Why? State because it is comparatively small.

- *"for Arctic atmospheric profiles retrieved from radiosonde measurements."*
  Please describe the source of these radiosondes measurements and give a few details.

- *"W threshold value after which $T_{b,j} \leq T_{b,k}$, or simply…"*
  A brief physical explanation of why this works would be helpful.

2.3

- *"The Melsheimer and Heygster (2008) algorithm extension is adapted only for sea ice surfaces."*
  Now would be an appropriate time to explain why they didn't apply it over ocean.

- *"From the data points over sea ice, the following regression relationship was found…"*
  Make it clear that this is from Melsheimer and Heygster (2008).

- *"The set of four parameters is determined through regression by using simulated brightness temperatures and atmospheric data from radiosonde profiles"*
  Some detail on how ARTS is used to do this would be helpful.

- *"for the L (low TWV), M (mid-TWV) and X (extended-TWV) cases. In the new algorithm, two extra sets of calibration parameters are required, for the M-ow (mid-TWV over open water) and X-ow (extended-TWV over open water) components."*
  These nicknames (e.g. "X-ow") are created but not used in the rest of the manuscript. Either used the shortened names or get rid of them entirely.

2.5

- *"from radiosondes profiles and simulated brightness temperatures."*
  Like was done previously? Please clarify.

2.6

- *"lead to differences in the third significant digit of the C0 and C1 parameters, which is small compared to other error contributions."*
  My interpretation of Fig. 3 is that $C(\tau_j, \tau_k)$ is only important for low values of TWV. But for those low values you're using an equation without $C(\tau_j, \tau_k)$, so is

it ever important? If not, why even bother with the term?

2.8

- "*One of the critical points in the algorithm...*"
- "*to the conditions used in their retrieval...*"
- "*the classical mid-TWV retrieval...*"
  As there are many algorithms discussed in this manuscript, it's confusing when you use vague references. Please be clear which algorithm you're referring to at all times.

- "*SSMIS or AMSR-E data...*"
  Need to define and cite both.

2.9

- Perhaps consider naming this section "Comparison of results to Melsheimer and Heygster (2008)", as "previous method" is, again, somewhat vague.

- "*The second dataset is the TWV product from Remote Sensing Systems...*"
  It would be helpful to mention that these algorithms only work over open ocean. It would also be helpful to give some basic information on the RSS and NN algorithms. What range of TWV can they retrieve? Do they cover the entire Artic? Do they work over sea or land ice (nope!)?

Results and Discussion

3.1

- "*Comparing this with the results in Fig. 5 shows that in the months where the contribution of the improved algorithm is greater, the correlation drop is more significant. Most of this contribution represents pixels with large TWV values, close to the retrieval limit that have a higher uncertainty.*"
  It would be valuable to compare the "original" algorithm and the "new" algorithm to ECMWF for only the pixels where they're both retrieving TWV. That way you could confirm that the poor correlations in June and September are primarily from the "new" algorithm retrieving over sea ice and open water in regions of high TWV. I also think it would be helpful to see a plot of the "new" retrieved TWV error as a function of the retrieved TWV. That is, quantify how the errors increase as you approach the saturation limit (~15 kg/m$^2$).

- "*the highest bias is again seen...*"
  Figure 6 just shows correlation, not necessarily a bias.

- "*For December there is an increase from −0.06 to −0.3 kg/m².*"
  This is only because you averaged them. The "Bias New" for all 3 Decembers is closer to zero than "Bias Original" but by averaging you came to the opposite conclusion.

- "*Earlier the algorithm was underestimating in both months compared to ECMWF by −0.38...*"
  June 2007 and June 2008 overestimate and June 2009 underestimates but by averaging you concluded that the algorithm was underestimating for all Junes.

- "*Thus, the increase of bias becomes highest in June.*"
  1.86 – (-0.38) = 2.24 and 1.29 – (-1.94) = 3.23 so the increase in bias is actually highest in September.

  Presumably the high bias in the "new" algorithm is due to the addition of retrievals over sea ice and open water. It would be helpful to actually show this. Additionally, do you have a physical explanation for the bias patterns seen in Fig. 7?

3.2

- "*with the former showing the lowest bias...*"
  Do you mean the latter? AMSU-B?

- "*While the AMSU-B method shows much higher negative bias values in summer it is important to note that average TWV values for the ice-free ocean in the summer months frequently surpass the saturation value of 15 kg/m².*"
  Is the correct interpretation here that your new AMSU-B algorithm is frequently observing scenes with TWV values of >15 kg/m² but still attempts a retrieval and gets values lower than 15 kg/m², resulting in a negative bias? Does this suggest that you need a better method than the one described in section 2 to prevent the algorithm from running on scenes that surpass the saturation value of TWV?

- "*Figure 9 top displays the average TWV...*"
  Might make more sense to show the top panel of Fig. 9 before Fig. 8.

- "*The average retrieved TWV for winter months matches better with the model while the overestimation for summer months, although still present is greatly diminished.*"
  Is an interpretation of Fig. 9 summer months that:
  - Over open ocean, often with high TWV, it does poorly (top panel)
  - The bottom panel shows that over all surfaces it does better
  - Thus, the bottom panel implies that over sea/land ice the algorithm must be doing very well compared to ECMWF?

If this is a logical interpretation, you should state as such and include a third panel in Fig. 9 of "New AMSU-B" plotted over ECMWF for only sea and land ice covered regions.

- "*This shows the difference between the average atmospheric water vapour load in the dry Central Arctic compared to the ice-free Arctic Ocean areas.*"
  My understanding is that it technically shows the difference between the entire Arctic and the ice-free Artic Ocean.

- "*The previous method was able to retrieve TWV over all surface types for atmospheric water vapour loads up to 6 kg/m² and over sea ice for up to 15 kg/m².*"
  This will be nice to see in a table near the beginning of the paper.

Conclusions

- "*(Fig. 9, Fig. 4)*"
  Do you mean Fig. 5 and Fig. 4?

- "*This difference can be explained by the additional area covered with the new algorithm.*"
  You could prove this by plotting the original and new AMSU-B vs. ECMWF for matched pixels.

- "*This demonstrates the capabilities of the method to retrieve TWV simultaneously over all surface types in the dry atmospheric conditions of the Arctic.*"
  Well, in the "dry" months you only get ~20% more data with the new algorithm and about the same bias as in the original algorithm. I think the stronger conclusion of this work is that the new algorithm provides greater spatial coverage, primarily in the warmer months, but that the new measurements, often at higher TWV values, have somewhat larger errors.

Figures:

- You capitalize "New" and "Original" in many of the figure captions but not in the manuscript. Please be consistent.

Fig. 3.

- What do the dashed horizontal lines represent?

Fig. 4

- Here you refer to the "new" algorithm as the "improved" algorithm. Please be consistent in your figures and the manuscript.

- Please make all the text larger.
- Please either make the plots larger or the land/ocean/country border lines thicker.
- The color for missing data (grey) doesn't contrast well with the high TWV color (white). Maybe consider upping the contrast somehow?

Fig. 6

- Dec. 2008 has a much better correlation than the other two Decembers for the "Original" algorithm. Any idea why?

Fig. 9

- Make all the text larger.
- Is the bottom panel open ocean and sea ice? Or open ocean, sea ice, and land ice, as stated in the manuscript?

Technical corrections:

P2 L5: "above-mentioned"
P2 L16: "radiation, and is…"
P3 L27: "fulfill"
P4 L6: "Infrared"
P4 L10: "Miao et al. (2001)"
P9 L8: "called the focal point"
P9 L18: New paragraph after "…regression fit."
P12 L8: "ocean/ice/land"
P12 L13: Don't start a new paragraph, as you're still talking about SEPOR/POLEX.
P14 L19: "of the AMSU-B instrument, sea surface temperature, and sea surface roughness…"
P15 L6: "(top panel of Fig. 2)
P15 L7: "following linear relationship in the form of Eq. 19"
P17 L11: "specific"
P18 L24: "Heygster (2008)"
P21 L5: Please cite the ECMWF ERA-Interim.
P22 L9: "new method (Fig. 5)."
P22 L24: "spatial contribution"
P23 L27: "closely follow"
P25 L11: Please define RMS.
P25 L15: "174%"
P26 L2: "high spatial coverage"

---

## Referee Comment (RC2) · Anonymous Referee #2 · 1 Dec 2017

The manuscript "Retrieval of Total Water Vapour in the Arctic Using Microwave Humidity Sounders" by Scarlat et al. (amt-2017-219) builds upon previously published algorithms for retrieving total (integrated) water vapour in the Arctic from TOA brightness temperature measurements by passive microwave radiometers on satellite platforms. Where the previous work was constrained to areas of high sea ice concentrations, the main purpose of the extended algorithm is to include areas over open ocean and variable sea ice. Such development is particularly important in light of decreasing sea ice extent in the Arctic. The manuscript is straightforward and well-organized, and the new approach generally well-documented and well-integrated into the previous work. However, I had difficulty interpreting the cross-validations in Section 3 and recommend that this be clarified or revisited in a revision. Also, there are many cumbersome sentences

and grammatical errors throughout the text (in particular near the beginning). I have flagged some examples under "Editorial Comments" and recommend the authors do a thorough edit for language before submitting the revision. The study is suitable for publication in Atmospheric Measurement Techniques, but my comments below should be addressed before publication.

General Comments:

I am struggling to interpret Section 3 for a number of reasons:

1) It was not clear that Fig. 5 was not actually a pixel count of the bottom panels on Fig 4, which according to the caption is just an example from 1 day. I think Figs 4 and 5 should be comparable. Perhaps keep the contents of Fig. 4 (the example is nice) and add additional panels for the "new" and "original" algorithms where instead of TWV, the pixels show the frequency of time that a retrieval is possible (perhaps for both January and July 2007-2009). This will provide the spatial context for Fig. 5.

2) You have not explained the domain over which you counted the pixels shown in Fig. 5. I'm assuming that this is the same as was plotted in Fig. 4 ($\sim$ north of 50°N) and if so it includes large regions of the north Atlantic and Pacific. It is likely then that a large number of newly-retrievable pixels are found outside the Arctic, which artificially inflates the percent increases reported in Section 3.1. Are you recommending that your method be applied in the north Atlantic and north Pacific too? If I understand correctly, the new algorithm is actually fairly limited in the more southerly parts of the map when averaged over time and these limitations are apparent also in Fig. 4. In addition to quantifying the improvement in coverage from the new algorithm, it would be interesting to know how close to total coverage is represented by the new algorithm. I recommend carefully defining the domain over which you recommend that algorithm be applied, explain this, and then use this domain to calculate all the results in Section 3.

3) I am confused about how to interpret the validation of the retrievals in comparison to

the ECMWF data. This is for two main reasons:

(3a) The ECMWF data are highly dependent on the model, especially at high latitudes, and are thus normally the type of data that is being validated. Therefore, it seems odd to conclude that the retrieval with the smallest systematic difference compared to ECMWF is the best retrieval. This is further confused by the fact that the ECMWF product likely assimilated the same/or similar data to that used in the retrievals. I realize you need a benchmark for comparison, so perhaps this can be resolved by tweaking the wording. What can you conclude through such a comparison?

(3b) AMSU-B new and AMSU-B orig are valid over different spatial areas and I assume the same is true for NN and RSS AMSR-E. I don't see where it is explained what spatial areas are averaged for the ECMWF data, but I know that it cannot apply to all four satellite algorithms at the same time. Therefore, it is not possible to isolate systematic differences in retrievals (which is interesting) to biases tied to spatial gaps (which was already established earlier in the section).

4) If the conclusion is that the new algorithm performs poorly compared to existing algorithms, what advantages are there to using it? If biases associated with spatial coverage could be separated from biases in the retrieval perhaps the advantages of the proposed algorithm would become clearer.

Specific Comments:

P4L19: AMSU-B is introduced here for the first time without explanation. I thought SSM/T2 was the data set being discussed.

P5L1-10: Can you more clearly articulate your motivation? While I learn later, it is not clear in the introduction why the Melsheimer and Heygster (2008) algorithm is unable to retrieve over the open ocean and marginal ice zone.

Section 2.8: What data set do you use to find the sea ice concentration?

P17L5-13: You have addressed the emissivity difference between 183 and 150 GHz

for the ocean component. Why not also develop an analogous correction for the sea ice so that this bias is corrected across the whole Arctic domain?

Editorial Comments:

P3L7-9: "Within this scenario . . ." is a very cumbersome sentence. Consider revising.

P3L11: no comma needed

P4L5: "Satellite retrievals also face. . ."

P4L22: "This assumption is false when switching" to "This is a poor assumption when using". The sentence is a bit odd anyway. Do you mean that the 183 triplet is used with 150 up to 2 kg/m2 after which one of the 183 bands is saturated and uncertainty increases?

P5L9: "allows for application of"

Fig 2: It would be better to label the panels as 2a and 2b and refer to them in the text accordingly.

P16L16: You mean for the channels used for retrievals in the mid-range of TWV?

P23L11: "method-specific"

P25L3: "improve the retrieval"

Figs 8 and 9: Please add "new" and "old" to the AMSU-B label in the legends.
* * *

---

## Author Comment (AC1) · 27 Dec 2017

We want to thank the anonymous referee for their work. We appreciate the direct and specific comments which address incomplete or unclear formulations on our part. Hopefully the answers below can address the referee's concerns and, after integration into a revised version of the manuscript can be accepted as satisfactory.

General comments

"1) It was not clear that Fig. 5 was not actually a pixel count of the bottom panels on Fig 4, which according to the caption is just an example from 1 day. I think Figs 4 and 5 should be comparable. Perhaps keep the contents of Fig. 4 (the example is nice) and add additional panels for the "new" and "original" algorithms where instead of TWV, the

pixels show the frequency of time that a retrieval is possible (perhaps for both January and July 2007-2009). This will provide the spatial context for Fig. 5."

This is a good suggestion. Figure 5 was supposed to show the difference in coverage area as a pixel count between the original and the new method for the whole temporal range we used (2007-2009). We will change Figure 4 to include the panels as you describe.

"2) You have not explained the domain over which you counted the pixels shown in Fig. 5. I'm assuming that this is the same as was plotted in Fig. 4 (âĹij north of 50âŮ̇ęN) and if so it includes large regions of the north Atlantic and Pacific. It is likely then that a large number of newly-retrievable pixels are found outside the Arctic, which artificially inflates the percent increases reported in Section 3.1. Are you recommending that your method be applied in the north Atlantic and north Pacific too? If I understand correctly, the new algorithm is actually fairly limited in the more southerly parts of the map when averaged over time and these limitations are apparent also in Fig. 4. In addition to quantifying the improvement in coverage from the new algorithm, it would be interesting to know how close to total coverage is represented by the new algorithm. I recommend carefully defining the domain over which you recommend that algorithm be applied, explain this, and then use this domain to calculate all the results in Section 3."

Both the original and the New method are only limited in their coverage by the water vapor content of the atmosphere. We apply them over the entire northern hemisphere at latitudes above 50 N, because the average atmospheric water vapor content at these latitudes is usually low enough for the methods to function. We will change the wording in the manuscript to reflect that the domain covers the entire polar region above 50 N, including the land masses of North America, Greenland and the northern regions of the Pacific and Atlantic Oceans.

"3) I am confused about how to interpret the validation of the retrievals in comparison

to the ECMWF data. This is for two main reasons: (3a) The ECMWF data are highly dependent on the model, especially at high latitudes, and are thus normally the type of data that is being validated. Therefore, it seems odd to conclude that the retrieval with the smallest systematic difference compared to ECMWF is the best retrieval. This is further confused by the fact that the ECMWF product likely assimilated the same/or similar data to that used in the retrievals. I realize you need a benchmark for comparison, so perhaps this can be resolved by tweaking the wording. What can you conclude through such a comparison?"

We agree that this comparison does not represent a validation because ECMWF is not an independent source for TWV data. We will change the wording to reflect that the testing done represents a comparison and not a validation effort for the presented methods. In this context the results show how well do the methods match with the benchmark of ECMWF TWV data which is often used in literature. The conclusion is that the Neural Network is the method that is closest to the model data when considering only the open water regions. Even though the AMSU-B method is not as close to the benchmark ECMWF data over open water, it is the only method that can also work over sea ice and land.

"(3b) AMSU-B new and AMSU-B orig are valid over different spatial areas and I assume the same is true for NN and RSS AMSR-E. I don't see where it is explained what spatial areas are averaged for the ECMWF data, but I know that it cannot apply to all four satellite algorithms at the same time. Therefore, it is not possible to isolate systematic differences in retrievals (which is interesting) to biases tied to spatial gaps (which was already established earlier in the section)"

For the inter-comparison in Figure 8 and Figure 9 top, we used the largest common domain for all three methods and collocated ECMWF data for this area. In essence the comparison for all methods against ECMWF is made over the ice-free open ocean where the RSS and Neural Network methods can work. This is why in the bottom panel of Figure 9 only the New AMSU-B retrieval mean value is compared to collocated ECMWF data because only this retrieval method and the model have valid data over all surface types (ocean, sea ice, land). Wherever the AMSR-E methods are discussed together with the AMSU-B retrievals the domain is the ice-free ocean. When the AMSU-B Original and New are compared to each other or to ECMWF, the domain is the entire Northern Hemisphere above 50N. The language in the revised version will clarify this aspect.

Specific comments:

"Section 2.8: What data set do you use to find the sea ice concentration?"

We used the ARTIST Sea Ice (ASI) algorithm which provides daily high resolution (6.25km) sea ice concentration data. The AMSU-B New TWV retrieval routine checks the ASI value for each pixel before deciding which module to use. We will add this information to the revised manuscript.

"P17L5-13: You have addressed the emissivity difference between 183 and 150 GHz for the ocean component. Why not also develop an analogous correction for the sea ice so that this bias is corrected across the whole Arctic domain?"

For the ocean surface there are reliable forward models which can be used to calculate the upward microwave emissivity, for example the RTTOV model which was used in this manuscript. Applying the same correction technique for sea ice would be difficult because the corresponding forward models for sea ice emissivity are not as mature as the open ocean models. Also, the nature of the variables required for parametrizing the sea ice microwave emissivity present an obstacle by themselves. Snow cover, sea ice type, snow water content are all parameters which are still difficult to obtain at our current level of development.

"P4L22: "This assumption is false when switching" to "This is a poor assumption when using". The sentence is a bit odd anyway. Do you mean that the 183 triplet is used with 150 up to 2 kg/m2 after which one of the 183 bands is saturated and uncertainty

increases?"

Yes, this is what we meant to express. We will edit that sentence accordingly.

"P16L16: You mean for the channels used for retrievals in the mid-range of TWV?"

Yes, mid-TWV refers to the channel selection that can retrieve in the range of 2-6 kg/mˆ2 which is the mid-range for the complete 0 to 14kg/mˆ2 of the method. The references to all modules and corresponding TWV ranges will be clarified by using one unitary naming convention (X – extended range, M – mid-range, L – low TWV range) and a table that shows which modules and parameters correspond to which range of retrievable TWV values.

We agree with all your other comments and we will implement them as is in the revised version of the manuscript.

---

## Author Comment (AC2) · 28 Dec 2017

General comments

We want to thank the anonymous referee for a very thorough and detailed review. If not especially addressed in our reply, we will implement these comments as is because we agree with the referee and we believe this is the simplest way to address those parts of the manuscript which were lacking in clarity.

In the following we try to address some specific comments and detail how we will modify the revised manuscript to satisfy the referee's concerns.

1) - "A table of all the algorithms would be immensely helpful."

[Figure]

This issue has been brought up in other discussions as well. The suggestion for a table to clarify the methods being used and their characteristic parameters is welcome and we will implement it in the revised manuscript.

Specific comments

2.6 - "My interpretation of Fig. 3 is that C($\tau$j, $\tau$k) is only important for low values of TWV. But for those low values you're using an equation without C($\tau$j, $\tau$k), so is it ever important? If not, why even bother with the term?"

This function has the largest variability in the low-TWV range of values where the retrieval equation does not include it. In the mid and extended range modules however this function is integrated in the retrieval equation and while it is not constant it varies slowly with increasing TWV. We show this behaviour in Figure 3 in order to support our assumption of using one constant value for the function. This constant value is different for the mid and extended range TWV retrieval modules. This clearly needs to be better explained and we will do so in the revised version.

2.9 - "It would be helpful to mention that these algorithms only work over open ocean. It would also be helpful to give some basic information on the RSS and NN algorithms. What range of TWV can they retrieve? Do they cover the entire Arctic? Do they work over sea or land ice (nope!)?"

Indeed this is one of the strongest points of our retrieval method and it needs to be better emphasized. Even though the RSS and NN algorithms show reliable results over open water, they are not suited for an Arctic wide retrieval which is the very motivation for our project.

3.1 - "It would be valuable to compare the "original" algorithm and the "new" algorithm to ECMWF for only the pixels where they're both retrieving TWV. That way you could confirm that the poor correlations in June and September are primarily from the "new" algorithm retrieving over sea ice and open water in regions of high TWV. I also think

it would be helpful to see a plot of the "new" retrieved TWV error as a function of the retrieved TWV. That is, quantify how the errors increase as you approach the saturation limit ($\sim$15 kg/m20)."

Comparing the New and Original versions only over common coverage areas was done internally and we have decided that the differences are very small and a plot would not bring sufficient new information to justify including it in the manuscript. This however seems to be an issue which touches on other unclear sections of the manuscript and including this plot might help bring the point across better. The areas covered by the Original algorithm are also covered by the New version using only slightly modified equations with the changes being detailed in Section 2.8. The main benefit of the New version when compared to the Original lies in the treatment of open water emissivities for the extended range which represents areas where the Original method simply couldn't retrieve anything.

- "Figure 6 just shows correlation, not necessarily a bias."

This part ("the highest bias is again seen...") is referring to Figure 7 which shows bias values between ECMWF and the two AMSU-B retrieval methods, and not Figure 6 which indeed shows correlation.

- "This is only because you averaged them. The "Bias New" for all 3 Decembers is closer to zero than "Bias Original" but by averaging you came to the opposite conclusion"

Regarding the interpretation of Figure 7 and the differences in bias between the two AMSU-B methods and the ECMWF benchmark, the errors you mention will be corrected. Indeed, using the 3 year average for the monthly bias values resulted in the wrong conclusions.

Considering that over the dry sea ice areas the two AMSU-B versions are almost identical the difference in bias vs ECMWF can only come from the new regions of open

water where the extended range TWV (6-14 kg/mˆ2) can be retrieved with the New algorithm. For the summer months (June and September), both the ratio of open water to sea ice cover is larger and the atmospheric water vapor load is higher than in winter. Both of these events will favor the use of the additional open water modules in the New algorithm. Looking at the comparison with RSS and NN methods that is done only over open water, this negative bias versus the ECMWF benchmark values is a particularity of the AMSU-B New algorithm. TWV values above the saturation limit will result in a negative brightness temperature difference ratio $\eta c$ and no numerical values can be retrieved in this case. As such only the retrieved values below the saturation threshold can be kept.

3.2 - "Do you mean the latter? AMSU-B?"

No, we mean the AMSR-E NN method which consistently shows the lowest magnitude bias throughout the time series.

- "Is the correct interpretation here that your new AMSU-B algorithm is frequently observing scenes with TWV values of >15 kg/m2 but still attempts a retrieval and gets values lower than 15 kg/m2 , resulting in a negative bias? Does this suggest that you need a better method than the one described in section 2 to prevent the algorithm from running on scenes that surpass the saturation value of TWV¿'

The 14-15 kg/mˆ2 is the upper limit above which numerical values cannot be returned from the retrieval equation. From an attempted retrieval in a scene with a ground truth TWV value ∼15 kg/m2, the method will retrieve the true value +- the retrieval error. From this retrieved value, only the underestimated values will be kept because values above this threshold cause Not a Number results from the retrieval equation. We are using this retrieval scheme up to the point where the retrieval equation fails and this inevitably means that there will be a negative bias close to this limit.

Fig. 3. - "What do the dashed horizontal lines represent?"

The dashed lines represent the range of variability for the C($\tau$j, $\tau$k) function for the corresponding TWV range. In the panel on the left this means the dashed lines cover the mid-range TWV domain (>2 kg/m2) while on the right panel they represent the extended range TWV domain (> 6kg/m2). We will add this explanation in the text.

Fig. 6 - "Dec. 2008 has a much better correlation than the other two Decembers for the "Original" algorithm. Any idea why?"

We are not sure why. Except for a small increase in the retrieval coverage for the Original method probably correlated to the larger sea ice extent for the winter season of 2008 there doesn't seem to be anything special about that year. We will check again for any mistakes in the correlation values.

Fig. 9 - "Is the bottom panel open ocean and sea ice? Or open ocean, sea ice, and land ice, as stated in the manuscript?"

It represents open ocean, sea ice and all land surfaces regardless of ice cover where the TWV value is low enough for the method to retrieve numerical values. The ice cover is relevant because it usually correlates with a drier atmosphere.
* * *

---

## Author Response (AR1)

The authors want to thank the anonymous referees for the thorough helpful comments. We hope we have been able to satisfy the concerns expressed in these comments and as a result the manuscript has been improved.

Below are the original comments together with our answers.
Colour coding:
- Comments of the referees
- Answers from the authors
- Line or figure in the revised manuscript addressing the comment

**Referee #1**

1) A table of all the algorithms would be immensely helpful.
We agree with this comment. We have implemented this table, together with accompanying references and explanations in the text.
P18 L11

2) It would be helpful to have a plot of new AMSU-B TWV vs. ECMWF as a function of TWV to show how the errors get worse as you approach the saturation limit of 15 kg/m2 .
Because we did not compare the retrieval with validation data we can only talk about the conclusions from comparing the retrieval with the ECMWF benchmark. The scatter of the retrieval increases when the TWV values increase and we have included a plot in Figure 9 which shows this behaviour. The discussion of this plot has been added to the manuscript.
P24 L12

3) Many of the equations presented in this manuscript are directly in Melsheimer and Heygster (2008) and it may not be necessary to duplicate them here.
In order to introduce the changes implemented in the new algorithm we have to discuss the module specific retrieval equations and we felt that the parameters required for these equations are also necessary (e.g. the calibration parameters, the C_tau functions). With the proper reference in place to the Miao et al. (2001) and Melsheimer and Heygster (2008) we feel that the equations are necessary for the sake of clarity.

4) The authors refer to an "original" algorithm multiple times throughout the manuscript. It should be clarified that this is referring specifically to the Melsheimer and Heygster (2008) algorithm.
We have added a sentence that states that the term "original algorithm" refers specifically to the Melsheimer and Heygster (2008) algorithm and have modified any other occurrence of the term in the text to avoid confusion.
P5 L27

Specific comments:

Abstract:
- You don't mention what you found. How did the new algorithm perform and what are you general conclusions? This needs to be in the abstract.
The conclusions about extended coverage in the warm months and at higher TWV values over open water with the connected higher uncertainties have been included in the abstract.
P2 L9

Introduction:

- "However this AMSU-B based method..."
Miao et al. (2001) uses SSM/T2. Regardless, please define acronyms the first time they are used.
Corrected the reference to AMSU-B and checked all other undefined acronyms in the text.

- "But the emerging errors were deemed acceptable as a trade-off for extending the retrieval range from 1.5-2 kg/m 2 (for only the three band channels) up to 7 kg/m 2 (for two band channels together with the 150 GHz channel)."
The use of "band channels" to refer to the 183 GHz channel specifically is confusing.
Included the exact frequencies of the channels being referred to in the text for clarity.
P5 L4

- "Melsheimer and Heygster (2008) extends the TWV retrieval range over sea ice by including the 89 GHz channel into the retrieval."
Include a short sentence on why including the 89 GHz channel is physically useful (it doesn't saturate as quickly).
Added a sentence that clarifies that the 183.31 +- 7 GHz channel is the one setting the limit for extended range TWV retrieval. The 89 GHz channel is used because it was next in line of the decreasing frequency channels.
P5 L8

- Much of the paragraph starting at P4 L9 feels like it belongs in section 2.2. Too many technical details for the introduction.
The technical details are also included in Section 2.2, however we feel that the introduction here of the channel frequencies is necessary for describing the state of the art for polar TWV retrieval. The limitations in retrieval range and spatial coverage are dictated by these technical details and the modifications brought on by the new method depend on these details.

- It may be useful to include a statement at the end of the introduction explaining what will be described in remaining sections (e.g. "Section 2.1 provides a discuss of the RT...").
Added such a paragraph which describes in short the structure of the manuscript.
P6 L4

Methods

2.1
- Need to make it clear early in the paper that you're primarily analysing AMSU-B measurements in this study.
Added one sentence that clarifies this aspect.
P6 L16

- "A down-looking microwave radiometer"
You may want to note that this is the same type of instrument as a "humidity sounder".
Added the note that AMSU-B is a humidity sounder.
P6 L14

2.2

- "In the original paper (Miao, 1998)..."
If this is the original paper that this work is based on, it deserves discussion in the introduction.
This reference points to the PhD dissertation of J. Miao where some of the aspects of TWV retrieval using the 183.31 band channels are discussed for the first time which is why we mention it in the Methods section, but the first retrieval over Antarctica is described in detail in Miao, (2001) which is the original Antarctic paper we discuss in the introduction.
The word "original" has been removed from the sentence to avoid confusion with the other Miao (Miao et al, 2001) paper which describes the Antarctic retrieval.

- "Because the Ts term is the same for both brightness temperatures, it has disappeared from Eq. (2) as a result of the subtraction."
Technically T s  is still in the b_ij  term and thus in Eq. (2).
That sentence was removed.

- "To find the relationship between the measured brightness temperature and the water vapour absorption we require the third brightness temperature measured in channel k."
A brief physical explanation of why three channels are necessary would be appreciated for people unfamiliar with microwave TWV retrievals.
Added a sentence that clarifies the use of the third brightness temperature for obtaining the ratio of brightness temperature differences which is a direct relationship between the measured Tbs and TWV.
P8 L14

- "Compared to the first two terms under the exponent, the quadratic term can be neglected..."
Why? State because it is comparatively small.
Added the clarification that the quadratic term is negligible small compared to the other two.

- "for Arctic atmospheric profiles retrieved from radiosonde measurements."
Please describe the source of these radiosondes measurements and give a few details.
The radiosonde data come from 27 WMO coastal and island stations in the Arctic, measured between 1996 and 2002. We added this information to the text.
 P10 L3

- "W threshold value after which T b,j  ≤ T b,k , or simply..."
A brief physical explanation of why this works would be helpful.
Added the physical explanation for for the Tb channel saturation switch.
 P11 L11

2.3

- "The Melsheimer and Heygster (2008) algorithm extension is adapted only for sea ice surfaces."
Now would be an appropriate time to explain why they didn't apply it over ocean.
The purpose of the Melsheimer and Heygster (2008) extension was to test this concept out for regions with relatively low TWV (sea ice covered regions) and sea ice surface emissivity data was readily accessible at the time for the required channels. While the use of radiative transfer software for open ocean emissivity simulations is mentioned as a future possibility in that original paper, it was not explored further at the time and we implemented it for this current work. This information has been

added to the text.
P13 L18

- "From the data points over sea ice, the following regression relationship was found..."
Make it clear that this is from Melsheimer and Heygster (2008).

Added a clear reference to the text.
P14 L17

- "The set of four parameters is determined through regression by using simulated brightness temperatures and atmospheric data from radiosonde profiles"
Some detail on how ARTS is used to do this would be helpful.
In Section 2.2 we describe the regression procedure for finding the focal point. With the focal point coordinates known we fit eq. (9) and retrieve the constant calibration parameters C_0 and C_1. Added reference to the proper section in the text.
P15 L14

- "for the L (low TWV), M (mid-TWV) and X (extended-TWV) cases. In the new algorithm, two extra sets of calibration parameters are required, for the M-ow (mid-TWV over open water) and X-ow (extended-TWV over open water) components."
These nicknames (e.g. "X-ow") are created but not used in the rest of the manuscript. Either used the shortened names or get rid of them entirely.
The shortened names had to be used in Table I, describing the different retrieval algorithms but besides that they are not used in the rest of the text. They are defined in the table caption and have been removed from the manuscript text.

2.5
- "from radiosondes profiles and simulated brightness temperatures."
Like was done previously? Please clarify.
This is the same procedure as used in Section 2.3 and initially described in Section 2.2. Because we are describing different sub-algorithms of the retrieval we wanted to emphasize that each module uses its own set of calibration parameters and all are derived using the same procedure (the one described in Section 2.2).
P15 L4

2.6
- "lead to differences in the third significant digit of the C0 and C1 parameters, which is small compared to other error contributions."
My interpretation of Fig. 3 is that $C(\tau j, \tau k)$ is only important for low values of TWV. But for those low values you're using an equation without $C(\tau j, \tau k)$, so is it ever important? If not, why even bother with the term?
This function has high variability in the low-TWV range of values where the retrieval equation does not include it. In the mid and extended range modules however this function is integrated in the retrieval equation and while it is not constant it does vary slowly with increasing TWV. We show this behavior in Figure 3 in order to back-up our assumption of using one constant value for the function. This constant value is different for the mid and extended range TWV retrieval modules respectively. We have rearranged the order of the sentences in order to clarify that this slow varying function is set to an approximate constant value individually calculated for the mid-TWV as was done for the extended-

TWV in the original Melsheimer and Heygster 2008 paper.
 P18 L7

"One of the critical points in the algorithm..."
"to the conditions used in their retrieval..."
"the classical mid-TWV retrieval..."
As there are many algorithms discussed in this manuscript, it's confusing when you use vague references. Please be clear which algorithm you're referring to at all times.
Added the convention that the very first TWV retrieval algorithm using humidity sounder measurements by Miao et al. (2001) is referred to as the Antarctic algorithm. The Melsheimer and Heygster (2008) algorithm is referred to as the original algorithm because it represents the original AMSU-B extended range retrieval method from which the new method, the focus of this current work, has been developed. The text has been modified throughout the manuscript to match this convention.

"SSMIS or AMSR-E data..."
Need to define and cite both.
This refers to the ASI sea ice data derived from either SSMIS or AMSR-E measurements. Added a clarification and the ASI reference to the text.
 P22 L16

Perhaps consider naming this section "Comparison of results to Melsheimer and Heygster (2008)", as "previous method" is, again, somewhat vague.
This section was meant to introduce the comparison work done using the new algorithm retrieval. This refers to comparison to the original AMSU-B method as well as the separate (over different spatial domains) inter-comparison with the two AMSR-E products (RSS and NN). The comparison with Melsheimer and Heygster (2008) is now Section 3. Section 2.9 was renamed to "Comparing the new retrieval with other TWV retrieval products" as all products and the spatial and temporal domain over which the comparison is done are introduced here.

"The second dataset is the TWV product from Remote Sensing Systems..."
It would be helpful to mention that these algorithms only work over open ocean. It would also be helpful to give some basic information on the RSS and  NN algorithms. What range of TWV can they retrieve? Do they cover the entire Arctic? Do they work over sea or land ice (nope!)?
Added extra details regarding the retrieval ranges and valid spatial domains for both algorithms.
 P23 L21

Results and Discussion

3.1

- "Comparing this with the results in Fig. 5 shows that in the months where the contribution of the improved algorithm is greater, the correlation drop is more significant. Most of this contribution represents pixels with large TWV values,  close to the retrieval limit that have a higher uncertainty."
It would be valuable to compare the "original" algorithm and the "new" algorithm to ECMWF for only the pixels where they're both retrieving TWV. That way you could confirm that the poor correlations in June and September are primarily from the "new" algorithm retrieving over sea ice and open water in regions of high TWV. I also think it would be helpful to see a plot of the "new" retrieved TWV error as a function of the retrieved TWV. That is, quantify how the errors increase as you approach the saturation limit (~15 kg/m $^2$ ).

Added three plots (Figure 8) which show the comparison of the two AMSU-B algorithms with each other, and each individually against ECMWF data. By comparing the two algorithms with each other, a plume of data points is visible where the TWV is overestimated by the new algorithm versus the original algorithm. This plume occurs in the extended range TWV domain, with values above 7 kg/m2. This overestimation of some of the data points by the new algorithm is confirmed by the comparison side by side of the new vs ECMWF and original vs ECMWF plots.

- "the highest bias is again seen..."
Figure 6 just shows correlation, not necessarily a bias.
This part is referring to Figure 10 in the revised manuscript which shows bias values between ECMWF and the two AMSU-B retrieval methods.

- "For December there is an increase from −0.06 to −0.3 kg/m 2 ."
This is only because you averaged them. The "Bias New" for all 3 Decembers is closer to zero than "Bias Original" but by averaging you came to the opposite conclusion.

- "Earlier the algorithm was underestimating in both months compared to ECMWF by −0.38..."
June 2007 and June 2008 overestimate and June 2009 underestimates but by averaging you concluded that the algorithm was underestimating for all Junes.

- "Thus, the increase of bias becomes highest in June."
1.86 – (-0.38) = 2.24 and 1.29 – (-1.94) = 3.23 so the increase in bias is actually highest in September.

- Presumably the high bias in the "new" algorithm is due to the addition of retrievals over sea ice and open water. It would be helpful to actually show this. Additionally, do you have a physical explanation for the bias patterns seen in Fig. 7?
All of these errors have been corrected and the discussion on the figure now refers to monthly values and not the averages over the three years.
P26 L28

The question of the high bias in the new algorithm is addressed by the plots in Figure 8 which show that because of the change in switching conditions (discussion in the text) between the original and new algorithm versions, there is a positive bias induced in the extended range TWV domain. The bias patterns in Figure 7 (Figure 10 in the revised version) are explained by the seasonal variability in ice cover and TWV range. In winter the new algorithm uses the dedicated open water modules and retrieves moderate TWV values with very low bias. In summer the original version used the extended range module for retrieving TWV over sea ice and it underestimated because values higher than 15 kg/m2 could be often times retrieved as below this value because of the proximity to the saturation limit. With the new algorithm, **moderately high TWV values (6-8 kg/m2) over open water that were retrieved by the mid-TWV module in the original algorithm have now been passed on to the extended range over open water module in the new algorithm** because of the stricter switch conditions. This module is responsible for the overestimation in summer in the new algorithm retrieved values. This is the only explanation that makes sense to us because this is the only difference between the original and the new algorithm in the spatial domain where both can retrieve valid results. The high TWV over open water regions could not be retrieved in the original algorithm so they are not part of this comparison.
The pixels that were retrieved over open water by the surface independent mid-TWV module at the limits of its sensitivity (because of the relaxed switch condition) are now retrieved by the higher uncertainty extended range open water module in the new algorithm. This more strict switch apparently

leads to lower bias in winter, but higher bias in summer. Because our target is the very low TWV in winter retrieval where no other reliable retrieval algorithms exist, we chose to keep this set up.

3.2

"with the former showing the lowest bias..."
Do you mean the latter? AMSU-B?
The former refers to the AMSR-E based NN approach which scores the lowest bias against ECMWF throughout the entire dataset over the common (all retrieval products) valid spatial domain.

"While the AMSU-B method shows much higher negative bias values in summer it is important to note that average TWV values for the ice-free ocean in the summer months frequently surpass the saturation value of 15 kg/m 2 ."
Is the correct interpretation here that your new AMSU-B algorithm is frequently observing scenes with TWV values of >15 kg/m 2  but still attempts a retrieval and gets values lower than 15 kg/m 2 , resulting in a negative bias? Does this suggest that you need a better method than the one described in section 2 to prevent the algorithm from running on scenes that surpass the saturation value of TWV?

The new AMSU-B algorithm is observing open water (the case for Figure 11) scenes with TWV values of 15  kg/m2 and attempts a retrieval. As the RMSD in this TWV domain is large (> 3  kg/m2) the values that are underestimated will be kept, the retrievals that could result in TWV values above 15 will fail the saturation cut off and will be discarded. Simply put, because of this hard cut off at 15 kg/m2, when the method retrieves values around 15 +- 3   kg/m2, only the underestimated values are retrieved as anything above the limit is discarded. In this sense we consider the switch conditions described in Section 2 as working correctly in moderately to high TWV domains. The disadvantage of this system is that high RMSD retrieval combined with the hard cut off at 15  kg/m2 results in a negative bias when there are many data points to be found in this TWV domain close to the saturation limit.

"Figure 9 top displays the average TWV..."
Might make more sense to show the top panel of Fig. 9 before Fig. 8.
We agree, however we want to show the comparison between the average values in the common valid spatial domain (Fig 12 top) with the existing AMSR-E based retrieval products (only open water) and the average TWV values over the spatial domain where the AMSU-B can retrieve valid data (open water + sea ice cover + land with low enough TWV – Fig. 12 bottom). This represents the key conclusion of this manuscript, that the new AMSU-B method can improve on the original algorithm capabilities for the dry Arctic areas and also provide higher coverage that can match the performance of existing data products in winter months, albeit with reduced performance in summer and in high TWV domains.

"The average retrieved TWV for winter months matches better with the model while the overestimation for summer months, although still present is greatly diminished."
Is an interpretation of Fig. 9 summer months that:

- Over open ocean, often with high TWV, it does poorly (top panel)

- The bottom panel shows that over all surfaces it does better

- Thus, the bottom panel implies that over sea/land ice the algorithm
must be doing very well compared to ECMWF? If this is a logical interpretation, you should state as

such and include a third panel in Fig. 9 of "New AMSU-B" plotted over ECMWF for only sea and land ice covered regions.

Yes this is a correct interpretation of the results we present, with the comment that over open water the new retrieval does poorly when TWV values are very high (for Arctic standards).
 A panel with AMSU-B retrieved TWV versus ECMWF TWV for only sea and land ice covered regions would mean the original AMSU-B algorithm results which are presented in Meslheimer and Heygster (2008). The improvements in the new algorithm are done for the retrieval over open water in the mid-TWV and extended range. As seen in Figure 9 (Figure 12 in the revised manuscript) over open water it matches the performance of the other retrieval products, and in Figure 7 (Figure 10 in the revised manuscript) in winter the new algorithm improves on the bias values for the original algorithm because of the use of the dedicated open water mid-TWV module.
- "This shows the difference between the average atmospheric water vapour load in the dry Central Arctic compared to the ice-free Arctic Ocean areas."
My understanding is that it technically shows the difference between the entire Arctic and the ice-free Arctic Ocean.
That is correct. We were referring to the big influence (the mean is dragged down) the dry sea ice covered regions of the Arctic have on the average TWV value calculated for the entire Arctic (including all ice free ocean regions above 50N). Changed the wording to better resemble the referee's comment.
 P29 L15

- "The previous method was able to retrieve TWV over all surface types for atmospheric water vapour loads up to 6 kg/m 2  and over sea ice for up to 15 kg/m 2 ."
This will be nice to see in a table near the beginning of the paper.
We have added this information to Table I.

Conclusions

- "(Fig. 9, Fig. 4)"
Do you mean Fig. 5 and Fig. 4?
Changed to Figures 4,5 and 12 of the revised manuscript.

- "This difference can be explained by the additional area covered with the new algorithm."
You could prove this by plotting the original and new AMSU-B vs. ECMWF for matched pixels.
This is now done in Figure 8.

- "This demonstrates the capabilities of the method to retrieve TWV simultaneously over all surface types in the dry atmospheric conditions of the Arctic."
Well, in the "dry" months you only get ~20% more data with the new algorithm and about the same bias as in the original algorithm. I think the stronger conclusion of this work is that the new algorithm provides greater spatial coverage, primarily in the warmer months, but that the new measurements, often at higher TWV values, have somewhat larger errors.
We agree, and we adapted the text to better match the conclusion of the Referee.

Figures:

- You capitalize "New" and "Original" in many of the figure captions but not in the manuscript. Please be consistent.
Eliminated all capitalized occurrences in the figure captions.

Fig. 3.

- What do the dashed horizontal lines represent?
The dashed horizontal lines represent the variability interval for the C(tau_j,tau_k) parameter inside the TWV range corresponding to each case. Added this sentence to the Figure 3 caption.

Fig. 4

- Here you refer to the "new" algorithm as the "improved" algorithm. Please be consistent in your figures and the manuscript.
Clarified all references to the new algorithm to conform with the naming convention.

- Please make all the text larger.

- Please either make the plots larger or the land/ocean/country border lines thicker.

- The colour for missing data (grey) doesn't contrast well with the high TWV colour (white). Maybe consider upping the contrast somehow?
In order to match the format of Figure 5, we added examples for the 1$^{st}$ of June and 1$^{st}$ of December 2009 for both original and new algorithms. The colour scale was changed to offer better contrast. Missing data is now white, and the high TWV areas are grey.
Figure 4

Fig. 6

- Dec. 2008 has a much better correlation than the other two Decembers for the "Original" algorithm. Any idea why?
We could not pinpoint any cause for this in the algorithm. This seems to be a natural occurrence specific to 2008 but we do not know why this happens.

Fig. 9

- Make all the text larger.
- Is the bottom panel open ocean and sea ice? Or open ocean, sea ice, and land ice, as stated in the manuscript?
It is open ocean, sea ice and any land surface where TWV value is low enough for the low or mid-TWV modules to work (i.e. below 7 kg/m2).
Figure 12 captions

Technical corrections:
We have implemented all of these technical corrections.

P2 L5: "above-mentioned"
P2 L16: "radiation, and is..."

P3 L27: "fulfill"
We have tried to use British spelling throughout the text.
P4 L6: "Infrared"
P4 L10: "Miao et al. (2001)"
P9 L8: "called the focal point"
P9 L18: New paragraph after "...regression fit."
P12 L8: "ocean/ice/land"
P12 L13: Don't start a new paragraph, as you're still talking about SEPOR/POLEX.
P14 L19: "of the AMSU-B instrument, sea surface temperature, and sea surface roughness..."
P15 L6: "(top panel of Fig. 2)
P15 L7: "following linear relationship in the form of Eq. 19"
P17 L11: "specific"
P18 L24: "Heygster (2008)"
P21 L5: Please cite the ECMWF ERA-Interim.
P22 L9: "new method (Fig. 5)."
P22 L24: "spatial contribution"
P23 L27: "closely follow"
P25 L11: Please define RMS.
P25 L15: "174%"
P26 L2: "high spatial coverage"

**Referee #2**

**General Comments:**

I am struggling to interpret Section 3 for a number of reasons:
1) It was not clear that Fig. 5 was not actually a pixel count of the bottom panels on Fig 4, which according to the caption is just an example from 1 day. I think Figs 4 and 5 should be comparable. Perhaps keep the contents of Fig. 4 (the example is nice) and add additional panels for the "new" and "original" algorithms where instead of TWV, the pixels show the frequency of time that a retrieval is possible (perhaps for both January and July 2007-2009). This will provide the spatial context for Fig. 5. Figure 5 was supposed to show the difference in coverage area as a pixel count between the original and the new method for the whole temporal range we used (2007-2009). We have introduced a new Figure 5 in the revised manuscript which shows the example coverage for the three years in December and June respectively for each of the two algorithms. The retrieval example in Figure 4 has been changed to match the format of Figure 5 with 4 maps, two for each method and for one day in December and one day in June. We agree that this graphical representation supports much better the point about the difference in spatial coverage between the new and original algorithm.

2) You have not explained the domain over which you counted the pixels shown in Fig. 5. I'm assuming that this is the same as was plotted in Fig. 4 ($\sim$ north of 50$\circ$ N) and if so it includes large regions of the north Atlantic and Pacific. It is likely then that a large number of newly-retrievable pixels are found outside the Arctic, which artificially inflates the percent increases reported in Section 3.1. Are you recommending that your method be applied in the north Atlantic and north Pacific too? If I understand correctly, the new algorithm is actually fairly limited in the more southerly parts of the map when averaged over time and these limitations are apparent also in Fig. 4. In addition to quantifying the improvement in coverage from the new algorithm, it would be interesting to know how close to total coverage is represented by the new algorithm. I recommend carefully defining the domain over which

you recommend that algorithm be applied, explain this, and then use this domain to calculate all the results in Section 3.

This comment is partially addressed by the new paragraph added for explaining the revised Figure 5. The domain is the entire Northern Hemisphere where the atmospheric water vapour values fall inside the  retrieval range of the algorithm. As an additional complexity, the surface type only matters for the extended range module. While below ~7 kg/m$^2$ the algorithm can retrieve TWV regardless of the surface type, above this relative threshold the algorithm can only function over open water or sea ice because of the specific reflectivity terms required by this module. This domain of "everything above 50$^\circ$N" has been used for all of the results in Section 3.
Added more details regarding the geographical domain used for the results in Section 3.
 P24 L21

3) I am confused about how to interpret the validation of the retrievals in comparison to the ECMWF data. This is for two main reasons:

> (3a) The ECMWF data are highly dependent on the model, especially at high latitudes, and are thus normally the type of data that is being validated. Therefore, it seems odd to conclude that the retrieval with the smallest systematic difference compared to ECMWF is the best retrieval. This is further confused by the fact that the ECMWF product likely assimilated the same/or similar data to that used in the retrievals. I realize you need a benchmark for comparison, so perhaps this can be resolved by tweaking the wording. What can you conclude through such a comparison?

We agree that ECMWF data does not represent validation level quality but it is a benchmark from which large deviations would signify problems with the retrieval. We see this comparison as a sanity test that could screen obvious deficiencies in the algorithm before more extensive and time consuming validation efforts are undertaken.
Added more details regarding why we chose ECMWF as a benchmark and what was our goal.
P28 L1

> (3b) AMSU-B new and AMSU-B orig are valid over different spatial areas and I assume the same is true for NN and RSS AMSR-E. I don't see where it is explained what spatial areas are averaged for the ECMWF data, but I know that it cannot apply to all four satellite algorithms at the same time. Therefore, it is not possible to isolate systematic differences in retrievals (which is interesting) to biases tied to spatial gaps (which was already established earlier in the section).

For each plot we always used the largest common spatial domain for all methods represented. For Figures 8 and 9 (top) this represents the common domain for all 3 retrieval products and corresponding ECMWF data. All sources were collocated so that the results shown represent the exact same data points coming from the three retrievals. The ECMWF average was calculated for the same number of pixels with collocated model data. This is why there are differences in the ECMWF column heights between Figure 9 top and bottom. The top represents only the open water areas where the AMSR-E based methods (RSS and NN) have valid values while the bottom represents all the areas where the full AMSU-B new algorithm works.
Clarified the methodology for comparing the retrieval products with ECMWF.
P28 L10

4) If the conclusion is that the new algorithm performs poorly compared to existing algorithms, what advantages are there to using it? If biases associated with spatial coverage could be separated from biases in the retrieval perhaps the advantages of the proposed algorithm would become clearer.
Biases associated with coverage are separated from biases in the retrieval. When comparing AMSU-B

new with RSS for example, we are only looking at the open ocean regions above 50◦ N. This means that we are comparing the RSS product with those modules of the AMSU-B retrieval that can function above the ice-free ocean and we can draw conclusions about the performance of these modules. The AMSU-B however is a collection of such modules and the output it produces is one consistent product that covers every region above 50◦ N where the TWV values are below the saturation limit. This includes landmasses where TWV is < ~8 kg/m2 and both sea ice covered and ice-free ocean regions where TWV is <~15 kg/m2. When comparing the new algorithm output with ECMWF data for the same spatial domain, the two agree much better than over open water only. As there are already proven retrieval products that can function over open water, the advantage of the new AMSU-B product lies in the fact that it can seamlessly retrieve over a much larger spatial domain which includes land and sea ice values up to 15 kg/m2 and with higher uncertainty over open water and when TWV values are towards the higher end of its retrieval interval.

Added details in the conclusions about the ranking of the algorithms and the strong points of the new AMSU-B retrieval.

P31 L2

Specific Comments:

P4L19: AMSU-B is introduced here for the first time without explanation. I thought SSM/T2 was the data set being discussed.

Clarified that SSM/T2 is the instrument used in Miao et al (2001) while this work is only based on AMSU-B measurements.

P6 L15

P5L1-10: Can you more clearly articulate your motivation? While I learn later, it is not clear in the introduction why the Melsheimer and Heygster (2008) algorithm is unable to retrieve over the open ocean and marginal ice zone.

The goal of the original paper was to provide an extended range TWV retrieval over sea ice as this was a capability that was missing before, while traditional passive microwave TWV retrievals over open water alone were an operational reality at the time. By only retrieving over high sea ice concentration regions however there are still gaps in the data coverage for the Arctic where the proximity of sea ice prevents open water algorithms to work, while the original AMSU-B algorithm could not retrieve in the extended TWV range if the ice concentration is lower than ~80%. This is where the current work on the new AMSU-B algorithm comes in to close this coverage gap and provide a retrieval solution that is consistent in time and space for the Arctic.

Added these extra details to the manuscript.

P5 L13

Section 2.8: What data set do you use to find the sea ice concentration?

We use collocated ARTIST Sea Ice concentration data (ASI) for deciding the application of open water/sea ice modules. This detail plus the reference was added to the text.

P20 L15

P17L5-13: You have addressed the emissivity difference between 183 and 150 GHz for the ocean component. Why not also develop an analogous correction for the sea ice so that this bias is corrected across the whole Arctic domain?

While the ocean emissivity is relatively well understood and possible to predict with models the treatment of sea ice surface emissivity is more complex. There has been one attempt at applying the emissivity difference correction over sea ice for the mid-range module by using sea ice emissivity data from the same SEPOR/POLEX campaign for the 183 and 150 GHz channels (Selbach, 2003). It was

deemed that the advantages of having one module that is surface independent in the mid-TWV range even with decreased accuracy over some ice surface types outweigh the complexity of implementing a correction for the emissivity difference between 183 and 150 GHz channels.

P3L7-9: "Within this scenario . . ." is a very cumbersome sentence. Consider revising.

We revised the sentence.
P3 L11

P3L11: no comma needed
Done

P4L5: "Satellite retrievals also face. . ."
Done

P4L22: "This assumption is false when switching" to "This is a poor assumption when using". The sentence is a bit odd anyway. Do you mean that the 183 triplet is used with 150 up to 2 kg/m2 after which one of the 183 bands is saturated and uncertainty increases?
We mean to say that the equal emissivity assumption is valid for the 183 triplet which can retrieve up to 2kg/m2. Above this value one of the 183 bands is saturated and needs to be replaced with the 150 Ghz channel. The new triplet will then use two 183 bands plus the 150 GHz channel and so the equal emissivity assumption is used even though we know that there is a difference in surface emissivities but the resulting increase in uncertainty is deemed acceptable for reasons described at point P17L5-13 above. The sentence was split and simplified.
P4 L26

P5L9: "allows for application of"
Done

Fig 2: It would be better to label the panels as 2a and 2b and refer to them in the text accordingly.
Done
P16L16: You mean for the channels used for retrievals in the mid-range of TWV?
Yes. We have clarified the sentence.

P23L11: "method-specific"
Done

P25L3: "improve the retrieval"
Done

Figs 8 and 9: Please add "new" and "old" to the AMSU-B label in the legends.
In these figures only the new algorithm is presented. The comparison with the two AMSR-E data products is done over open water only and the original AMSU-B algorithm could not retrieve TWV with values above 7 kg/m2 over open water. This would have restricted the common valid spatial domain very much as such low values are encountered in close proximity to sea ice, which is where the two AMSR-E algorithms would not have any valid data.
Specified in the figure captions as well as in the text which spatial domain was used for the comparison.

[revised manuscript text omitted]